# La RoSA: Enhancing LLM Efficiency via **La**yerwise **Ro**tated **S**parse **A**ctivation

Kai Liu [* 1]   Bowen Xu [* 1]   Shaoyu Wu [1]   Xin Chen [1]   Hao Zhou [1]   Yongliang Tao [1]   Lulu Hu [1]

## Abstract

Activation sparsity can reduce the computational overhead and memory transfers during the forward pass of Large Language Model (LLM) inference. Existing methods face limitations, either demanding time-consuming recovery training that hinders real-world adoption, or relying on empirical magnitude-based pruning, which causes fluctuating sparsity and unstable inference speedup. This paper introduces LaRoSA (**La**yerwise **Ro**tated **S**parse **A**ctivation), a novel method for activation sparsification designed to improve LLM efficiency without requiring additional training or magnitude-based pruning. We leverage layerwise orthogonal rotations to transform input activations into rotated forms that are more suitable for sparsification. By employing a Top-K selection approach within the rotated activations, we achieve consistent model-level sparsity and reliable wall-clock time speed-up. LaRoSA is effective across various sizes and types of LLMs, demonstrating minimal performance degradation and robust inference acceleration. Specifically, for LLaMA2-7B at 40% sparsity, LaRoSA achieves a mere 0.17 perplexity gap with a consistent 1.30× wall-clock time speed-up, and reduces the accuracy gap in zero-shot tasks compared to the dense model to just 0.54%, while surpassing TEAL by 1.77% and CATS by 17.14%.

## 1. Introduction

Large Language Models (LLMs) have achieved significant advancements across various real-world machine learning applications and leaderboards (DeepSeek-AI, 2025; OpenAI, 2024; Team, 2024; Qwen et al., 2025). However, billions of parameters in these models pose significant challenges for efficient LLM inference and serving, especially in terms of memory footprint and computational overhead. Re-

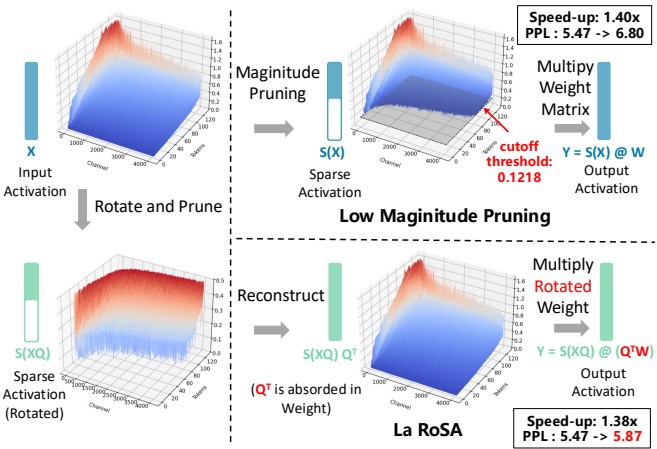

Figure 1. **LaRoSA** rotate and prune the input activations to achieve more consistent and efficient LLM inference. The rotation is orthogonal and can be reversed after pruning. The overall method provide a more accurate approximation of the full computation. Visualizations are from the 12th Attention block of LLaMA2-7B.

cent efforts on boosting LLM inference have predominantly focused on low-bit model quantization (Frantar et al., 2022; Lin et al., 2024b; Ashkboos et al., 2024b; Liu et al., 2024b; Lin et al., 2024a) and model weights pruning methods (Frantar & Alistarh, 2023; Ashkboos et al., 2024a; Ma et al., 2023; Fang et al., 2024). Such techniques enhance inference speed and save memory usage by either employing low-precision parameter formats or reducing the total number of parameters. Exploiting activation sparsity (Kurtz et al., 2020; Li et al., 2022; Chen et al., 2023) is another promising approach to improve inference efficiency. Since channels in the weight matrix corresponding to zero-valued activations are not utilized during the forward pass, LLM inference can be accelerated by: **(1) avoiding transfer such parameters between memory and computation units**, and **(2) omitting the matrix computations associated with those weights channels.** These speed-ups could be achieved by utilizing sparse patterns that are compatible with hardware.

Recent studies (Mirzadeh et al., 2024; Liu et al., 2023; Zhang et al., 2024) have observed a high degree of activation sparsity in the ReLU-based Transformers. Based on this, Dejavu (Liu et al., 2023) trains a predictor for each layer to preemptively identify weight channels corresponding to zero-valued activations. (Mirzadeh et al., 2024) induce

*Equal contribution  [1]Alibaba Group. Correspondence to: Kai Liu <kane.liukai@gmail.com>, Bowen Xu <bowiehsu1024@gmail.com>.

*Proceedings of the 42nd International Conference on Machine Learning*, Vancouver, Canada. PMLR 267, 2025. Copyright 2025 by the author(s).

high sparsity by replacing non-ReLU activations with ReLU. However, these methods do not achieve full sparsity in the models and require substantial additional training to recover model performance. Some other works have turned their attention to magnitude-based pruning. CATS (Lee et al., 2024) utilizes empirical distribution histogram to obtain the cutoff threshold in MLP blocks, and zero out activations below that threshold. TEAL (Liu et al., 2024a) further extends this concept to all input activations in both Attention and MLP blocks. However, offline cutoff thresholds are heavily related to calibration tokens, and lack accuracy and consistence in generating optimal sparse patterns for unseen tokens to achieve a target sparsity level. Moreover, inconsistent activation sparsity can lead to unstable inference speed-ups, reducing efficiency in LLM inference. Driven by these limitations, we raise the following question: *Is it possible to **quickly, effectively, and consistently** leverage activation sparsity to enhance the efficiency of LLMs without additional training or empirical thresholds?*

In this work, we present LaRoSA (**La**yerwise **Ro**tated **S**parse **A**ctivation), a training-free activation sparsification method that produces optimal sparse patterns and consistent generation speed-up for LLM inference. As depicted in Figure 1, our method uses an orthogonal rotation matrix $\mathbf{Q}$ to transform the input activation, enabling more effective activation sparsification. We apply PCA to obtain the rotation matrix $\mathbf{Q}$ for each layer and employ a Top-K function to acquire stable activation sparsity. A residual adapter is added to realize layer-specific independent rotations. In the end, the rotation matrix $\mathbf{Q}$ can be absorbed by the corresponding weight matrix, eliminating the additional computational overhead introduced by $\mathbf{Q}$. Our main contributions are:

- We show that layerwise orthogonal rotation matrices can be applied to improve the performance and efficiency of LLMs without extra training or empirical thresholds.

- We introduce consistent model-level sparsity to non-ReLU based LLMs by applying Top-K functions on the rotated activations and realize more stable and efficient model inference than SOTA methods.

- We conduct extensive experiments with leading LLMs and demonstrate that LaRoSA is effective and robust across different types, sizes, and sparsity levels. LaRoSA presents minimal performance degradation while providing consistent wall-clock time speed-up.

## 2. Related Work

**Activation Sparsity** Activation sparsity refers to the phenomenon where a considerable portion of a model's hidden states are zero-valued. This property naturally emerges in the intermediate states of ReLU-based MLP blocks (Li et al., 2022). Liu et al. (2023) utilized activation sparsity

to accelerate LLM decoding speed by omitting the transfer of weight channels associated with zero-valued entries to GPU registers. Furthermore, Song et al. (2023) and Alizadeh et al. (2024) leveraged activation sparsity to facilitate CPU offloading, significantly reducing memory transfer overhead between CPUs and GPUs. However, it is noteworthy that modern leading LLMs (Team, 2024; Qwen et al., 2025) utilize more complex nonlinear functions (e.g., SwiGLU (Shazeer, 2020)) which do not inherently produce sparsity (Mirzadeh et al., 2024). Consequently, most of the ReLU-based activation sparsification methods are not applicable to these models.

**Inducing Sparsity in non-ReLU Models** Mirzadeh et al. (2024) replaced SwiGLU with ReLU in LLMs, followed by continued pre-training to induce activation sparsity. Song et al. (2025) and Song et al. (2024) introduced techniques such as activation regularization to further enhance sparsity in adapted models. Zhang et al. (2024) experimented with various activation functions and identified Squared ReLU (So et al., 2021) as the most effective alternative. CATS (Lee et al., 2024) realizes a desired sparsity level in SwiGLU based MLP output activations through empirical distribution and magnitude thresholding. TEAL (Liu et al., 2024a) further extends this concept to the input activations of self-attention and MLP blocks, and prunes low-magnitude activations. Wang et al. (2024) combines magnitude pruning and quantized activations and presents scaling laws for sparsely activated LLMs. However, these methods either rely on substantial post-training and calibration, or produce unstable model-level sparsity, resulting in sparse activated LLMs that lack efficiency and flexibility.

**Rotation-based Model Compression** Recent works have focused on using orthogonal rotation matrices to mitigate performance drops in LLM pruning and quantization. SliceGPT (Ashkboos et al., 2024a) multiplies each weight matrix by an orthogonal matrix to reduce the embedding dimension, and use a residual adapter to achieve computational invariance. QuaRot (Ashkboos et al., 2024b) use random Hadamard rotation to redistribute outliers across all channels. SpinQuant (Liu et al., 2024b) suggests training the orthogonal matrix instead of using a random Hadamard matrix to further enhance quantizability.

## 3. Preliminary

### 3.1. Activation Sparsity in LLMs

In this work, we focus on the Transformer architecture LLMs which are composed of repeated decoder layers (Touvron et al., 2023). In subsequent sections, unless otherwise specified, a 'layer' refers to the decoder layer that repeatedly appears after the token embedding layer, and a 'block' refers to an Attention block or a MLP block from a decoder layer.

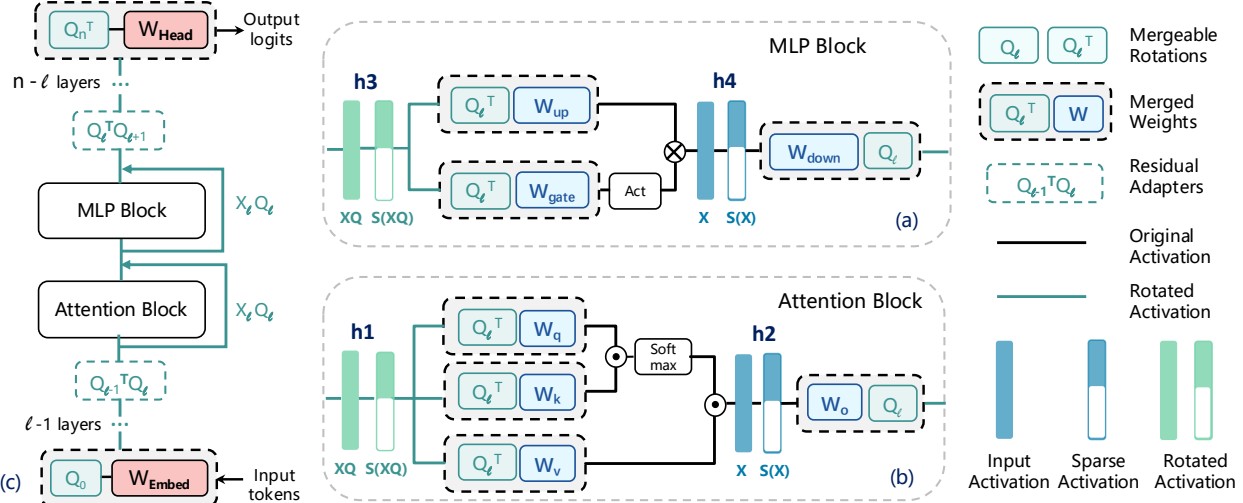

*Figure 2.* Model inference with LaRoSA. **(a)&(b)** The layerwise orthogonal matrix $\mathbf{Q}_l$ can be absorbed into weight matrices to avoid extra computations, ensuring that the input activations for each block are pre-transformed by the layer-specific orthogonal matrix $\mathbf{Q}_l$. **(c)** Introducing residual adapters in the residual stream ensures that each layer's input activations have independent orthogonal rotations. The $\mathbf{Q}_0$ of first layer and $\mathbf{Q}_n$ of last layer can be merged into the weight matrix of token embedding and LM head layer, respectively.

We assume that the network is constructed using a pre-norm approach, meaning that a LayerNorm or RMSNorm operation precedes each block. For clarity, $\boldsymbol{h}_1$ and $\boldsymbol{h}_2$ refer to the Attention block input activations, and $\boldsymbol{h}_3$ and $\boldsymbol{h}_4$ to the MLP block input activations. Figure 2 illustrates the construction of such blocks and layers. Typically, the forward pass in each block is performed by linear mappings between input tensors and weight matrices, which can be expressed as:

$$\mathbf{Y} = \mathbf{X} \cdot \mathbf{W}^T, \tag{1}$$

where $\mathbf{X} \in \mathbb{R}^{N \times D_{\text{in}}}$ denote the input tensor of $N$ tokens, $\mathbf{W} \in \mathbb{R}^{D_{\text{in}} \times D_{\text{out}}}$ is the weight matrix, and $\mathbf{Y} \in \mathbb{R}^{N \times D_{\text{out}}}$ is the output tensor. $D_{\text{in}}$ and $D_{\text{out}}$ are hidden sizes of input and output tensor. The matrix multiplication shown in Equation (1) is identical for each $\mathbf{W}_q$, $\mathbf{W}_k$, $\mathbf{W}_v$, $\mathbf{W}_o$ in Attention blocks and $\mathbf{W}_{\text{up}}$, $\mathbf{W}_{\text{gate}}$, $\mathbf{W}_{\text{down}}$ in MLP blocks. Let $\boldsymbol{x} \in \mathbb{R}^{D_{\text{in}}}$ denotes the input activation of a single token from $\mathbf{X}$. The activation sparsity $p \in [0, 1]$ of $\boldsymbol{x}$ is defined as the proportion of zero-valued elements:

$$p = \frac{1}{D} \sum_{i=1}^{D} \mathbf{1}(\boldsymbol{x}_i = 0), \tag{2}$$

where $x_i$ is the i-th element of $\boldsymbol{x}$, and the indicator function $\mathbf{1}(\boldsymbol{x}_i = 0)$ outputs 1 if the activation value is equal to zero and 0 otherwise. As highlighted in (Mirzadeh et al., 2024), non-ReLU based LLMs inherently exhibit extremely low activation sparsity. Recent studies (Lee et al., 2024; Liu et al., 2024a) have assumed that low-magnitude activations exert minimal effects on model outputs and have shown that magnitude-based pruning is empirically effective. Typically,

the activation sparsity ins these methods is defined as:

$$p = \frac{1}{D} \sum_{i=1}^{D} \mathbf{1}(\boldsymbol{x}_i \leqslant \epsilon), \tag{3}$$

where $\epsilon$ is a cutoff threshold estimated from the empirical cumulative distribution of activations, obtained offline from calibration datasets. The linear mappings in magnitude pruning method is then formulated by:

$$\mathbf{Y} = S_\epsilon(\mathbf{X}) \cdot \mathbf{W}^T, \tag{4}$$

where $S_\epsilon(\cdot)$ is the sparsification function defined as:

$$S_\epsilon(\boldsymbol{x}_i) = \begin{cases} 0, & \text{if } |\boldsymbol{x}_i| \leq \epsilon \\ \boldsymbol{x}_i, & \text{otherwise,} \end{cases} \tag{5}$$

### 3.2. Computational Invariance Transformation

The computational invariance theorem, as described in (Ashkboos et al., 2024a) Theorem 1, asserts that the activations between Transformer blocks (i.e., Attention and MLP) can be transformed using an orthogonal matrix $\mathbf{Q}$ without altering the model's output. This theorem remains valid even when RMSNorm is applied between the blocks. The reason is that RMSNorm divides activations by their norms, and a orthogonal transformation $\mathbf{Q}$ does not affect these norms, since $\mathbf{Q}\mathbf{Q}^\top = \mathbf{Q}^\top\mathbf{Q} = \mathbf{I}$ and $\|\boldsymbol{x}\mathbf{Q}\| = \sqrt{\boldsymbol{x}\mathbf{Q}\mathbf{Q}^\top\boldsymbol{x}^\top} = \sqrt{\boldsymbol{x}\boldsymbol{x}^\top} = \|\boldsymbol{x}\|$. Therefore, we have the the commutation property in RMSNorm operation:

$$\text{RMSNorm}(\mathbf{X}) = \text{RMSNorm}(\mathbf{X}\mathbf{Q})\mathbf{Q}^\top. \tag{6}$$

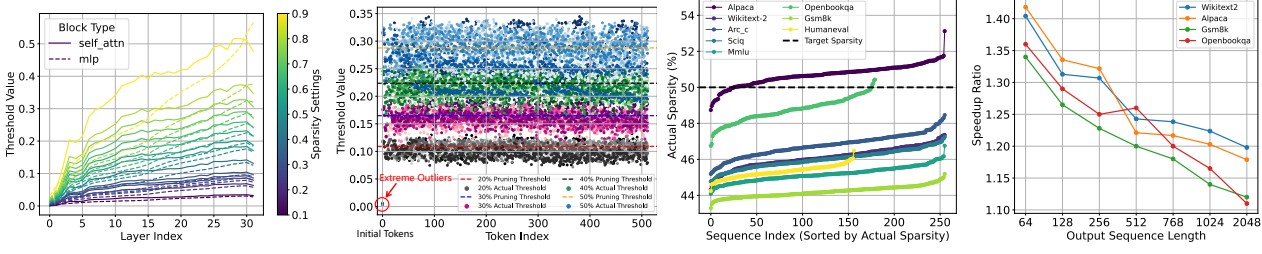

(a) Magnitude Pruning Thresholds    (b) Actual Thresholds    (c) Inconsistent Actual Sparsity    (d) Unstable Inference Speedup

*Figure 3.* **(a)** Offline calibrated thresholds for Attention and MLP blocks of LLaMA3-8B. **(b)** The calibrated and actual needed thresholds in 24th Attention block of LLaMA3-8B. Input tokens are randomly selected from WikiText2. The dashed lines denote the calibrated thresholds, while scatter points indicate real thresholds needed to achieve the target sparsity. Many blue points (50% actual thresholds) are even below the black line (40% calibrated threshold). **(c)** Average actual activation sparsity of LLaMA3-8B's Attention blocks. Input sequences are from different datasets and sorted by the sparsity. **(d)** The inference speedup of magnitude pruning at 50% sparsity under various output sequence lengths. Speedup equal to 1.0 is the dense version of LLaMA3-8B model. Best view in color and zoom in.

In forward pass, we can merge the additional transformation $\mathbf{Q}$ into weight matrices to avoid the extra computation induced by $\mathbf{Q}$. As depicted in Figure 2, if $\mathbf{W}_o$ is right-multiplied by an orthogonal matrix $\mathbf{Q}$, this will make the Attention block produces a rotated activation $\mathbf{XQ}$ as output. We can cancel this effect by left-multiplying the weight matrices of subsequent MLP blocks (i.e., $\mathbf{W}_{up}$, $\mathbf{W}_{gate}$) by $\mathbf{Q}^\top$, so the output changes induced by $\mathbf{Q}$ are neutralized, and the original computation results in each block remain unchanged. This process is same for $\mathbf{W}_{down}$ and $\mathbf{W}_q$, $\mathbf{W}_k$, $\mathbf{W}_v$.

## 4. Method

### 4.1. Motivation: Rethinking Low-Magnitude Pruning

We conduct experiments on LLaMA3-8B using magnitude pruning methods, using 16 sequences of 512 tokens from WikiText2 (Merity et al., 2016) for various sparsity settings. We identify several key observations and visualizing them in Figure 3.

**Ambiguity and Inaccuracy in Defining Low Magnitude**. The definition of low magnitude influences how activations are sparsified. As shown in Figure 3a, cutoff thresholds obtained from offline calibration fluctuate significantly as the layer goes deeper. The thresholds for Attention and MLP blocks within the same layer also exhibit great variation across different sparsity levels. Using these inaccurate thresholds can lead to severe mismatches with target sparsity. Figure 3b compares cutoff thresholds for activation pruning using TEAL and TopK methods at various sparsity levels. TEAL uses a fixed threshold from offline calibration, shown as a dashed line, while TopK computes thresholds during testing, shown as scatter points. It's evident that offline calibrated thresholds hardly align with the actual needed thresholds. In most cases, tokens actually require higher or lower thresholds than the calibrated ones. At sequence

beginnings, actual thresholds are extremely lower than calibrated ones, explaining why methods like TEAL struggle to sparsify initial tokens of a sequence during prefill stage of LLM inference.

**Difficulty in Maintaining Consistent Sparsity**. Achieving stable and predictable LLM inference acceleration requires maintaining uniform and consistent target sparsity across different input sequences. As illustrated in Figure 3c, conventional pruning approaches that use calibrated cutoff thresholds fail to deliver precise sparsity control, resulting in actual sparsity significantly deviating from target values and showing considerable variation across sequences from different datasets. These inconsistencies directly compromise the stable acceleration results shown in Figure 3d, leading to inefficient LLM inference that falls short of fully realizing the theoretical performance advantages of sparse activation. This observation highlights the critical importance of consistent sparsity for two key reasons: **1)** Inconsistent sparsity prevents models from reliably achieving their target sparsity, making computational efficiency gains unpredictable and rendering optimization efforts futile. **2)** It introduces substantial variance in per-token inference times, causing undesirable fluctuations in decoding speed.

**Magnitude and Channel Importance Misconception**. Magnitude pruning methods (Liu et al., 2024a; Lee et al., 2024) typically assume that low-magnitude activations have a minor impact on model outputs, leading to the practice of zeroing out entries that fall below a certain threshold. However, as proved in Wanda (Sun et al., 2023) and SliceGPT (Ashkboos et al., 2024a), pruning solely based on the magnitude of weights or activations is not optimal. Low-magnitude activations can still substantially affect the output if their corresponding channels in the weight matrix have large norms. We further demonstrate in Appendix A that empirical output errors from magnitude pruning methods are consistently higher than theoretical output errors.

## 4.2. Layerwise Orthogonal Rotation

The limitations of magnitude pruning drive us to explore a more effective approach for identifying activations that are inherently easier to prune. Inspired by rotation-based quantization methods (Ashkboos et al., 2024b), we attempt to obtain a rotated activation through orthogonal transformations, after which the channel importance can be easily distinguished. We propose using an orthogonal matrix $\mathbf{Q}_l$ to rotate and prune the input activation (i.e., $\boldsymbol{h}_1$ and $\boldsymbol{h}_3$) of $l$-th layer, and reverse the rotation using $\mathbf{Q}_l^\top$ after pruning. To construct $\mathbf{Q}_l$, we use PCA to leverage it's inherent ability to identify principal components. Specifically, we select a calibration dataset consist of $M$ sequences, feed it into the model and forward pass through each layer. Let $\mathbf{X}_l^i$ denotes the input activation of layer $l$ for $i$-th sequence. We compute the covariance matrix $\mathbf{X}_l^i(\mathbf{X}_l^i)^T$ of each $\mathbf{X}_l^i$ and average over all sequences:

$$Cov(\mathbf{X}_l, \mathbf{X}_l^T) = \frac{1}{M} \sum_{i=0}^{M} \mathbf{X}_l^i(\mathbf{X}_l^i)^T. \tag{7}$$

Then, the eigenvectors of $Cov(\mathbf{X}_l, \mathbf{X}_l^T)$ are solved and sorted in eigenvalues descending order to construct the $\mathbf{Q}_l$. Here, we only rotate $\boldsymbol{h}_1$ and $\boldsymbol{h}_3$ in LaRoSA for efficiency, since rotation matrices for $\boldsymbol{h}_2$ and $\boldsymbol{h}_4$ cannot be incorporated into $\mathbf{W}_\text{v}$, $\mathbf{W}_\text{up}$, and $\mathbf{W}_\text{gate}$ due to the constraints imposed by the Grouped Query Attention (GQA) in Attention block and the element-wise product operation in MLP block.

It can be observed that each layer must use the same $\mathbf{Q}_l$ due to residual connections between layers. We initially attempt to use a single global $\mathbf{Q}$ for all layers, but quickly find out that the optimal orthogonal matrix for each layer tends to be very different. To address this, a residual adapter $\mathbf{Q}_l^T\mathbf{Q}_{l+1}$ is applied to the output of each layer to realize layer-specific rotation, as shown in Figure 2. Although this introduce small additional operations, we demonstrate in Section 5.3 that such operations preserve the optimal rotation for each layer and improve the overall efficiency. In Appendix A, we provide a theoretical analysis of rotation-based activation pruning, demonstrating that the constructed rotation matrix effectively reduces empirical error within each layer, thereby outperforming magnitude pruning methods in accuracy.

## 4.3. Introducing Consistent Activation Sparsity

To introduce a consistent activation sparsity into non-ReLU LLMs, we replace magnitude pruning with Top-K function to zero out the rotated activations. Specifically, given the rotated activation $\boldsymbol{x}_l\mathbf{Q}_l$ from the $l$-th block and a target sparsity level $p$, we first calculate the number of elements $k$ that need to be reserved after sparsification by $k = \alpha \cdot (1 - p) \cdot D_\text{in}$, where $\alpha$ is a hyperparameter tuned to control the sparsity coefficient between $\boldsymbol{h}_1$ and $\boldsymbol{h}_2$ (or $\boldsymbol{h}_3$ and $\boldsymbol{h}_4$) within the same block and guarantee the overall model-level

sparsity. We use Top-K function to determine whether each element of rotated activation $\tilde{\boldsymbol{x}} = \boldsymbol{x}\mathbf{Q}$ need to be reserved or not. The sparsification function in LaRoSA is defined as:

$$S_k(\tilde{\boldsymbol{x}}_i) = \begin{cases} \tilde{\boldsymbol{x}}_i, & \text{if } |\tilde{\boldsymbol{x}}_i| \in \text{Top}_k(|\tilde{\boldsymbol{x}}_i|) \\ 0, & \text{otherwise} \end{cases} \tag{8}$$

where $\text{Top}_k$ is the function that selects the largest $k$ elements of $\tilde{\boldsymbol{x}}$ in terms of the absolute values. Activation sparsity can be introduced by modifying the forward pass to:

$$\mathbf{Y}_l = S_k(\mathbf{X}_l\mathbf{Q}_l) \cdot \mathbf{Q}_l^T\mathbf{W}_l^T, \tag{9}$$

where $\mathbf{Q}_l^T\mathbf{W}_l^T$ can be precomputed and merged into the weights to avoid online computation. Therefore, Equation (9) can be rewritten as :

$$\mathbf{Y}_l = S_k(\mathbf{X}_l\mathbf{Q}_l) \cdot (\mathbf{W}_l\mathbf{Q}_l)^T, \tag{10}$$

and $\mathbf{Q}_l$ become identity matrix $\mathbf{I}$ when activation $\boldsymbol{x}$ is in the $\boldsymbol{h}_2$ and $\boldsymbol{h}_4$ position for efficient implementation.

## 4.4. Hardware-efficient Customized Kernels

Introducing activation sparsity alone is insufficient to achieve end-to-end LLM inference acceleration on hardware (Frantar & Alistarh, 2023). To translate the activation sparsity induced by LaRoSA into wall-clock inference speed-up, we need to implement a fast and efficient customized GPU kernel. Building on the Triton-based kernel introduced by DejaVu (Liu et al., 2023) and TEAL (Liu et al., 2024a), we implement a GEMV kernel that provides practical and consistent token generation speed-up. **1)** We store weight matrices $\mathbf{W}$ in column-major format in memory to facilitate the selective loading of weight columns corresponding to non-zero activations, as memory coalescing could significantly improve memory bandwidth utilization (Dao et al., 2022; Ivanov et al., 2020). **2)** We fuse the Top-K function into the matrix-vector multiplication, identifying the indices of the columns in the weights that need to be retained during the forward pass. **3)** We selectively load the sparse activations and corresponding weight columns from memory, transfer them to the computation units, perform the multiplication, and store the result.

## 5. Experiments

### 5.1. Experimental Setups

**Models and Baselines**. To test the robustness and scalability of LaRoSA, we conducted experiments using 4 leading LLMs: LLaMA3 (Team, 2024), LLaMA2 (Touvron et al., 2023), Mistral (Jiang et al., 2023), and Qwen2.5 (Qwen et al., 2025). The model size is selected from 7B and 70B parameters. We compare with state-of-the-art (SOTA) baselines, including CATS (Lee et al., 2024) and TEAL (Liu

*Table 1.* Comparison of the average accuracy on seven zero-shot tasks and 5-shot MMLU. The first column indicates the actual model-level sparsity. CATS cannot achieve 50% model-level sparsity as it only sparsifies the MLP's Down projections. Full results are in Appendix C.

| Method | LLaMA2-7B | | LLaMA2-70B | | LLaMA3-8B | | LLaMA3-70B | | Qwen2.5-7B | | Qwen2.5-72B | | Mistral-7B | |
|---|---|---|---|---|---|---|---|---|---|---|---|---|---|---|
| | $Acc_7$ 0-shot | MMLU 5-shot | $Acc_7$ 0-shot | MMLU 5-shot | $Acc_7$ 0-shot | MMLU 5-shot | $Acc_7$ 0-shot | MMLU 5-shot | $Acc_7$ 0-shot | MMLU 5-shot | $Acc_7$ 0-shot | MMLU 5-shot | $Acc_7$ 0-shot | MMLU 5-shot |
| Dense | 66.69 | 45.85 | 73.66 | 68.80 | 70.05 | 65.26 | 76.29 | 78.71 | 70.34 | 74.21 | 75.58 | 86.08 | 70.44 | 62.34 |
| CATS$_{25\%}$ | 63.71 | 42.76 | 72.74 | 67.50 | 67.66 | 61.85 | 75.58 | 77.93 | 69.66 | 72.67 | 72.77 | 84.91 | 64.55 | 59.83 |
| TEAL$_{25\%}$ | 65.94 | 44.66 | 73.31 | 67.70 | 69.40 | 63.85 | 73.23 | 74.86 | 69.76 | 73.21 | 75.05 | 85.44 | 70.06 | 61.51 |
| LaRoSA$_{25\%}$ | **66.39** | **45.66** | **73.38** | **68.74** | **69.54** | **64.85** | **76.30** | **78.13** | **70.12** | **73.74** | **75.53** | **85.62** | **70.25** | **61.81** |
| CATS$_{40\%}$ | 49.55 | 24.67 | 62.74 | 55.83 | 55.11 | 31.82 | 72.07 | 72.12 | 61.83 | 63.99 | 63.12 | 80.95 | 59.62 | 44.31 |
| TEAL$_{40\%}$ | 64.92 | 43.46 | 72.47 | 66.78 | 68.14 | 59.84 | 72.24 | 73.23 | 68.61 | 71.44 | 74.65 | 84.80 | 68.76 | 60.17 |
| LaRoSA$_{40\%}$ | **66.15** | **44.66** | **73.31** | **68.16** | **68.79** | **62.61** | **75.41** | **77.62** | **69.67** | **72.33** | **75.35** | **85.53** | **69.44** | **61.15** |
| TEAL$_{50\%}$ | 63.22 | 39.57 | 71.92 | 64.43 | 64.92 | 52.78 | 70.80 | 69.20 | 67.76 | 68.53 | 73.74 | 83.54 | 66.73 | 57.34 |
| LaRoSA$_{50\%}$ | **64.61** | **43.10** | **72.86** | **67.57** | **67.19** | **58.65** | **73.81** | **76.51** | **69.09** | **70.09** | **75.18** | **84.34** | **68.46** | **58.80** |

*Table 2.* Perplexity results under different sparsity settings. The numbers in the first column reflects the actual model-level sparsity

| Method | LLaMA3 | | LLaMA2 | | Mistral | Qwen2.5 | |
|---|---|---|---|---|---|---|---|
| | **8B** | **70B** | **7B** | **70B** | **7B** | **7B** | **72B** |
| Dense | 6.13 | 2.85 | 5.47 | 3.32 | 5.31 | 6.85 | 3.87 |
| CATS$_{25\%}$ | 7.22 | 3.56 | 5.99 | 3.81 | 6.38 | 7.58 | 4.74 |
| TEAL$_{25\%}$ | 6.37 | 3.95 | 6.31 | 3.43 | 5.53 | 6.93 | 3.93 |
| LaRoSA$_{25\%}$ | **6.23** | **2.94** | **5.51** | **3.34** | **5.34** | **6.90** | **3.92** |
| CATS$_{40\%}$ | 18.37 | 5.97 | 45.46 | 10.75 | 23.44 | 11.06 | 7.15 |
| TEAL$_{40\%}$ | 6.83 | 4.45 | 6.40 | 3.61 | 5.98 | 7.20 | 4.07 |
| LaRoSA$_{40\%}$ | **6.60** | **3.37** | **5.64** | **3.44** | **5.44** | **7.10** | **4.06** |
| TEAL$_{50\%}$ | 7.56 | 5.61 | 6.80 | 3.91 | 7.17 | 7.81 | 4.31 |
| LaRoSA$_{50\%}$ | **7.22** | **4.10** | **5.87** | **3.62** | **5.62** | **7.42** | **4.26** |
| TEAL$_{60\%}$ | 9.19 | 9.97 | 7.82 | 4.53 | 8.05 | 9.99 | 7.45 |
| LaRoSA$_{60\%}$ | **8.57** | **5.51** | **6.40** | **3.98** | **6.04** | **8.42** | **4.90** |

et al., 2024a). For SOTA baselines, we choose not to quote the results from their papers in order to make fair comparisons under the same model-level sparsity setting and to impose full sparsity in the input sequence length dimension.

**Implementation Details.** We use the WikiText2 train set (Merity et al., 2016) as calibration dataset for LaRoSA and other reproducible works. We randomly select 16 sequences with sequence length of 2048 tokens to compute the rotation matrices **Q** for LaRoSA and empirical distributions for CATS and TEAL. The computation of **Q** is performed on 8×80G A100 GPUs, taking approximately 12 minutes to complete for the LLaMA3 70B model. For sparsity coefficient $\alpha$, we employ Grid Search to find the optimal hyperparameter for each model, as shown in Appendix B Algorithm 1. The optimal $\alpha$ for each activation type of models is presented in Appendix B Table 11. For a fair comparison, we implement full activation sparsification at prefill stage for all methods including TEAL.

**Evaluation.** We report the perplexity (PPL) score on Wiki-Text2 test set for generation task. All models are evalu-

ated on the same 128 random samples with a 2048-token context length. For comprehensive assessment, we utilize lm-evaluation-harness(Gao et al., 2023) to assess LaRoSA performance on a variety of downstream tasks, including 0-shot tasks for ARC-Easy (Clark et al., 2018), ARC Challenge (Clark et al., 2018), PIQA (Bisk et al., 2020), BoolQ (Clark et al., 2019), HellaSwag (Zellers et al., 2019), OpenBookQA (Mihaylov et al., 2018) and WinoGrande (Sakaguchi et al., 2021). We also report 5-shot accuracy for MMLU (Hendrycks et al., 2020).

### 5.2. Main Results

**Language Generation Task.** First, we evaluate the accuracy of LaRoSA with SOTA baselines on language generation task. As indicated in Table 2, LaRoSA notably outperforms other baselines on perplexity across various models and different sparsity settings. Under a low sparsity setting like 25%, LaRoSA achieves a minimal PPL loss of merely 0.1 on LLaMA3-70B. This significantly surpasses previous methods, such as CATS with a PPL loss of 0.71 and TEAL with a PPL loss of 1.1. LaRoSA maintains its advantage even at high sparsity levels, such as 60%. For instance, on Qwen2.5-72B, LaRoSA incurs only a 1.03 PPL loss, considerably outperforming TEAL, which suffers a loss of 3.87. Moreover, on LLaMA2-7B, LaRoSA at 50% sparsity even outperforms the previous SOTA methods under a 25% sparsity setting, highlighting its superior efficiency. In TEAL's paper, they only sparsify 99% of tokens in the prefill stage because the empirical thresholds they use struggle to sparsify initial tokens. In contrast, LaRoSA does not suffer this performance degradation and can achieve full sparsity in the input sequence length dimension.

**Zero-shot and Few-shot Tasks.** LaRoSA exhibits superior performance when applied to downstream language tasks. We present the end-to-end performance of LaRoSA and SOTA methods across different models, tasks, and sparsity ratios. The results are shown in Table 1. We highlight sev-

*Table 3.* Single-batch token generation speed (token/sec) results of LaRoSA. We exclude LLaMA2-70B since it is architecturally similar to to LLaMA3-70B. Tensor parallelism (TP2) is set for LLaMA3-70B and Qwen2.5-72B.

| GPU | Sparsity | LLaMA2 7B | LLaMA3 8B | LLaMA3 70B | Qwen2.5 7B | Qwen2.5 72B | Mistral 7B |
|---|---|---|---|---|---|---|---|
| | Dense | 109.95 (1.00×) | 101.25 (1.00×) | 22.05 (1.00×) | 107.54 (1.00×) | 19.32 (1.00×) | 105.88 (1.00×) |
| A100 | 0% | 96.75 (0.88×) | 92.13 (0.89×) | 20.32 (0.90×) | 100.97 (0.90×) | 56.33 (0.92×) | 56.33 (0.88×) |
| | 25% | 125.34 (1.14×) | 111.37 (1.10×) | 24.70 (1.12×) | 123.67 (1.15×) | 22.60 (1.17×) | 118.59 (1.12×) |
| | 50% | 151.73 (1.38×) | 131.62 (1.30×) | 29.77 (1.35×) | 152.71 (1.42×) | 29.17 (1.51×) | 148.23 (1.40×) |
| | 75% | 188.01 (1.72×) | 174.15 (1.72×) | 38.59 (1.75×) | 180.67 (1.68×) | 33.42 (1.73×) | 186.35 (1.76×) |
| | Dense | 72.47 (1.00×) | 67.19 (1.00×) | 15.75 (1.00×) | 71.70 (1.00×) | 14.80 (1.00×) | 72.89 (1.00×) |
| H20 | 0% | 65.95 (0.91×) | 60.47 (0.90×) | 14.33 (0.91×) | 63.81 (0.89×) | 13.62 (0.92×) | 72.89 (0.90×) |
| | 25% | 85.51 (1.18×) | 79.28 (1.18×) | 19.21 (1.22×) | 83.17 (1.16×) | 18.35 (1.24×) | 65.60 (1.16×) |
| | 50% | 104.36 (1.44×) | 94.74 (1.41×) | 22.84 (1.45×) | 98.95 (1.38×) | 21.46 (1.45×) | 84.55 (1.40×) |
| | 75% | 136.97 (1.89×) | 126.99 (1.89×) | 29.92 (1.90×) | 132.64 (1.85×) | 28.42 (1.92×) | 102.05 (1.90×) |

*Table 4.* Accuracy of math and knowledge tasks on reasoning models under different sparsity settings.

| Method | MATH-500 | GPQA-Diamond | AIME-2024 |
|---|---|---|---|
| **DS-R1-Distill-Llama3-8B** | | | |
| DeepSeek-R1 Report | 89.1 | 49.0 | 50.4 |
| Reproduce Result | 87.6 | 45.9 | 40.0 |
| +LaRoSA 25% | 85.0 | 44.9 | 40.0 |
| +LaRoSA 40% | 80.6 | 43.1 | 36.7 |
| +LaRoSA 50% | 78.8 | 41.2 | 33.3 |
| **DS-R1-Distill-Qwen2.5-7B** | | | |
| DeepSeek-R1 Report | 92.8 | 49.1 | 55.5 |
| Reproduce Result | 93.4 | 43.9 | 50.0 |
| +LaRoSA 25% | 91.0 | 43.5 | 46.7 |
| +LaRoSA 40% | 88.5 | 41.0 | 43.3 |
| +LaRoSA 50% | 85.1 | 40.2 | 43.3 |

eral key observations: **1**) LaRoSA significantly outperforms CATS and TEAL across zero-shot tasks and 5-shot MMLU. With the same 40% model-level sparsity, LaRoSA achieves an average performance gain of 16.60% over CATS and 1.23% over TEAL on LLaMA2-7B. **2**) LaRoSA achieves performance comparable to the dense model with negligible degradation at a low sparsity level around 25%. For example, LaRoSA shows only a 0.22% accuracy drop on Qwen2.5-7B compared to the dense model. **3**) For some models, such as LLaMA2-7B, LaRoSA at 40% sparsity performs even better than CATS and TEAL do at 25% sparsity. These results demonstrate the superiority of our LaRoSA, which establishes new state-of-the-art performance by effectively sparsifying input activations.

**Complex Problem Solving Tasks**. To further evaluate the effectiveness and robustness of LaRoSA, we conduct experiments on complex tasks such as MATH500 (Lightman et al., 2024), GPQA-Diamond (Rein et al., 2024), and AIME'24 (AIME, 2025) using two reasoning models:

DeepSeek-R1-Distill-Llama3-8B and DeepSeek-R1-Distill-Qwen2.5-7B (DeepSeek-AI, 2025). We utilize the Alpaca dataset, which is not inherently related to complex reasoning tasks, to derive the rotation matrix $\mathbf{Q}_l$ for each layer. The evaluation of LaRoSA on these complex tasks is carried out with the Hugging Face Open-R1 repository (Face, 2025). As shown in Table 4, our experimental results indicate that at a sparsity level of 25%, both reasoning models exhibit only a slight performance degradation on complex tasks. This finding demonstrates that the LaRoSA method maintains its effectiveness even when utilizing data that is not directly related to complex reasoning tasks for generating the rotation matrix.

**Inference Speed Up**. We demonstrate consistent end-to-end efficiency improvements with LaRoSA. For this, we collected ten samples, each consisting of 128 tokens, from various test datasets and generated new sequences with lengths ranging from 128 to 2048 tokens. This was done to evaluate performance across different generation lengths. We applied 50% sparsity to LLaMA3-8B and Qwen2.5-7B using optimal sparse patterns obtained from LaRoSA and TEAL. As depicted in Figure 4, our LaRoSA realizes more consistent wall-clock time speed-ups compared to TEAL. This improvement is attributed to the TopK strategy which precisely zeros out activation entries needed for a specific sparsity level. As the output sequence length increases, LaRoSA outperforms TEAL in overall speed-up efficiency, demonstrating its effectiveness and robustness. Moreover, we evaluated the end-to-end token generation speed-up on different types of models at a fixed output sequence length of 128, with results detailed in Table 3. For LLaMA3-70B, LaRoSA achieves notable speed-ups of up to 1.45× and 1.90× at 50% and 75% sparsity, respectively. The speed-up for 0% sparsity is lower than dense model, due to the additional computation introduced by residual adapters and sparsification functions. Nonetheless, we observe overall inference efficiency improvements as sparsity increases. The

*Table 5.* Impact of different sparsification functions on model's actual sparsity and perplexity results. Experiments are performed on LLaMA2-7B. Sparsity is set to 50%.

| Method | Attention | | MLP | | Model | PPL |
|---|---|---|---|---|---|---|
| | $h_1(\%)$ | $h_2(\%)$ | $h_3(\%)$ | $h_4(\%)$ | Sparsity | |
| TEAL | $47.9_{(\pm1.2)}$ | $52.0_{(\pm4.7)}$ | $48.4_{(\pm1.4)}$ | $54.9_{(\pm1.5)}$ | 48.8 | 6.80 |
| TopK | $50.0_{(\pm0.0)}$ | $50.0_{(\pm0.0)}$ | $50.0_{(\pm0.0)}$ | $50.0_{(\pm0.0)}$ | 50.0 | 6.25 |
| TopK+GS | $45.0_{(\pm0.0)}$ | $65.0_{(\pm0.0)}$ | $40.0_{(\pm0.0)}$ | $57.5_{(\pm0.0)}$ | 50.0 | 6.02 |
| LaRoSA | $45.0_{(\pm0.0)}$ | $65.0_{(\pm0.0)}$ | $40.0_{(\pm0.0)}$ | $57.5_{(\pm0.0)}$ | 50.0 | **5.87** |

*Table 6.* Influence of different rotation types in various sparsity settings. **Baseline method represent TopK+GS**.

| Method | LLaMA3-8B | | LLaMA2-7B | |
|---|---|---|---|---|
| | PPL | TFLOPs | PPL | TFLOPs |
| Dense | 6.13 | 32.94 (1.00×) | 5.47 | 29.26 (1.00×) |
| Baseline$_{25\%}$ | 6.29 | 32.94 (1.00×) | 5.53 | 29.26 (1.00×) |
| Baseline$_{25\%}$ + $\mathbf{Q}_M$ | 6.35 | 32.94 (1.00×) | 5.64 | 29.26 (1.00×) |
| Baseline$_{25\%}$ + $\mathbf{Q}_B$ | 6.23 | 37.22 (1.13×) | 5.51 | 33.06 (1.15×) |
| Baseline$_{25\%}$ + $\mathbf{Q}_L$ | **6.23** | 34.91 (1.06×) | **5.51** | 31.46 (1.07×) |
| Baseline$_{50\%}$ | 7.71 | 32.94 (1.00×) | 6.02 | 29.26 (1.00×) |
| Baseline$_{50\%}$ + $\mathbf{Q}_M$ | 7.76 | 32.94 (1.00×) | 6.03 | 29.26 (1.00×) |
| Baseline$_{50\%}$ + $\mathbf{Q}_B$ | **7.21** | 37.22 (1.13×) | **5.86** | 33.06 (1.15×) |
| Baseline$_{50\%}$ + $\mathbf{Q}_L$ | 7.22 | 34.91 (1.06×) | 5.87 | 31.46 (1.07×) |
| Baseline$_{60\%}$ | 9.39 | 32.94 (1.00×) | 6.91 | 29.26 (1.00×) |
| Baseline$_{60\%}$ + $\mathbf{Q}_M$ | 9.43 | 32.94 (1.00×) | 6.92 | 29.26 (1.00×) |
| Baseline$_{60\%}$ + $\mathbf{Q}_B$ | **8.56** | 37.22 (1.13×) | 6.40 | 33.06(1.15×) |
| Baseline$_{60\%}$ + $\mathbf{Q}_L$ | 8.57 | 34.91( 1.06×) | **6.40** | 31.46 (1.07×) |

detailed extra computation in TFLOPS are listed in Table 19.

**Impact of Initial Tokens**. As noted in Section 5.4.3 of the TEAL paper, TEAL applies activation sparsification to only 99% of tokens during the prefill stage. This approach is attributed to attention sinks (Xiao et al., 2024), which cause more severe degradation when sparsifying the initial tokens. Consequently, TEAL skips the first 1% of tokens to maintain model performance. However, we argue that this issue arises due to a mismatch between the actual and calibrated thresholds for each token. As illustrated in Figure 3b, this mismatch is particularly pronounced at the beginning of sequences, leading to TEAL's suboptimal performance on initial tokens. In contrast, LaRoSA and CATS apply sparsification uniformly across all tokens without special treatment. For a fair comparison, we implement full activation sparsification for TEAL, which results in slightly worse performance than originally reported by TEAL. This outcome also underscores the lack of robustness in using empirical thresholds.

### 5.3. Ablation Study

**Impact of Sparsification Function**. We ablate four activation sparsification functions in LaRoSA: **1) TEAL** refers to the function that uses a fixed cutoff threshold to prune low-magnitude entries, as described in Equation (5). **2) TopK**

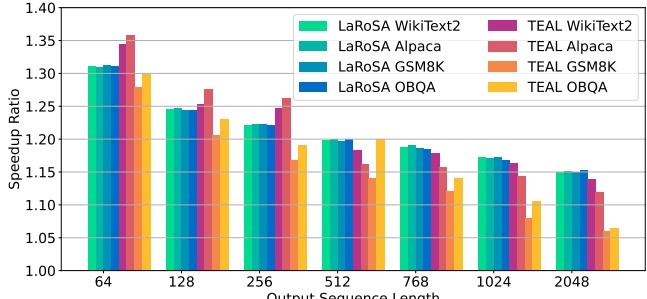

(a) Inference Speed-up for LLaMA3-8B.

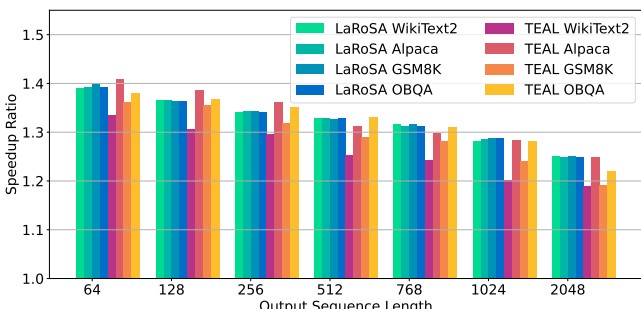

(b) Inference Speed-up for Qwen2.5-7B.

*Figure 4.* Comparison on inference speed-ups at 50% sparsity. Experiments are conducted on NVIDIA A100 GPUs.

denotes the approach of introducing sparsity directly using the TopK function. **3) TopK+GS** indicates an enhancement of TopK by incorporating Grid Search to obtain optimal coefficients $\alpha$ for different inputs within the same block. **4) LaRoSA** represents the complete method, which introduces a layerwise rotation matrix $\mathbf{Q}_l$ on top of TopK+GS. Table 5 shows that the actual sparsity of the TEAL exhibits fluctuations and fails to achieve a 50% model-level sparsity. In contrast, TopK function can consistently maintain the actual sparsity of each input activation around 50% and significantly reducing PPL. By introducing the optimal sparsity coefficient $\alpha$ found through Grid Search, the PPL loss can be further minimized while keep the overall model sparsity remains unchanged. Finally, incorporating layerwise orthogonal rotation matrices further enhance model's performance.

**Model-level vs. Layer-level vs. Block-level Rotation**. We comprehensively investigate the number and sources of rotation matrices and their impact on the performance and efficiency of sparsified models. Three types of rotation matrices are analyzed: **1) Model-level rotation**: This approach averages the covariance matrices of input across all layers to obtain a single and shared $\mathbf{Q}_M$ for all layers, without adding residual adapters. **2) Block-level rotation**: Each Attention and MLP block in a layer receives its own rotation matrix $\mathbf{Q}_B$, derived from its input's covariance matrix. Thus, a model with $L$ layers generates $2L$ rotation matrices and $2L$ residual adapters. **3) Layer-level rotation**: Used by LaRoSA, this strategy assigns a unique rotation matrix $\mathbf{Q}_L$

*Table 7.* Impact of different calibration datasets. The samples are used for calculating the covariance matrices for input activations.

| Method | Calibration Dataset | LLaMA3 8B | | LLaMA2 7B | |
|---|---|---|---|---|---|
| | | PPL($\downarrow$) | Acc$_7$($\uparrow$) | PPL($\downarrow$) | Acc$_7$($\uparrow$) |
| Dense | - | 6.13 | 70.05 | 5.47 | 66.69 |
| LaRoSA$_{25\%}$ | WikiText2 | 6.13 | 69.54 | 5.51 | 66.39 |
| LaRoSA$_{25\%}$ | Alpaca | 6.13 | 69.87 | 5.52 | 66.36 |
| LaRoSA$_{40\%}$ | WikiText2 | 6.60 | 68.79 | 5.64 | 66.15 |
| LaRoSA$_{40\%}$ | Alpaca | 6.61 | 68.69 | 5.72 | 66.33 |
| LaRoSA$_{50\%}$ | WikiText2 | 7.22 | 67.19 | 5.87 | 64.61 |
| LaRoSA$_{50\%}$ | Alpaca | 7.23 | 67.60 | 5.94 | 64.85 |

*Table 8.* Influence of calibration datasets on reasoning models. Cos.Simi. is obtained by averaging the cosine similarities of the covariance matrices across layers when using different calibration data. Experiments are performed on DS-R1-Distill-Qwen2.5-7B.

| Method | Calibration Dataset | Cos. Simi. | 0-shot Acc$_7$ | MATH-500 | GPQA-Diamond |
|---|---|---|---|---|---|
| LaRoSA$_{25\%}$ | Alpaca | 1.0 | 82.9 | 91.0 | 43.5 |
| LaRoSA$_{25\%}$ | WikiText2 | 0.998 | 83.0 | 91.0 | 43.7 |
| LaRoSA$_{25\%}$ | C4 | 0.997 | 82.9 | 91.2 | 43.5 |

to each layer based on its input covariance matrix, resulting in $L$ rotation matrices and $L$ residual adapters. As shown in Table 6, these rotation types result in different additional computation overhead and distinct improvements in PPL. Although $\mathbf{Q}_M$ does not introduce additional overhead, it actually results in worse performance compared to the baseline. Distinct $\mathbf{Q}_B$ for each block significantly increases computational overhead, yet offers only marginal advantages over $\mathbf{Q}_L$. Consequently, the layerwise rotation matrix $\mathbf{Q}_L$ emerges as the optimal choice, as it increases limited computations with notable performance enhancements.

**Influence of Calibration Datasets**. We apply LaRoSA to sparsify the LLaMA2-7B and LLaMA3-8B models using different calibration datasets (i.e. WikiText2 and Alpaca (Taori et al., 2023)), with results shown in Table 7. It can be observed that the choice of calibration datasets has a relatively minimal effect on sparsified model performance. This is due to our method utilize calibration data solely for calculating covariance matrices of each layer's input activation, rather than for empirical distribution computation as employed by methods like CATS and TEAL. Additionally, we validate the performance of LaRoSA within the reasoning model on complex tasks. As shown in Table 8, the covariance matrices of input activations within the same layer exhibit a high degree of cosine similarity across different data inputs. This observation suggests that the acquisition of layerwise rotation matrices is not strongly correlated with the calibration data. The performance in zero-shot and more complex tasks further demonstrates that the calibration data has minimal impact on the model's per-

*Table 9.* Compatiblity of LaRoSA with 4-bit weight quantization.

| Precision | Quant. Methods | LLaMA3-8B | | LLaMA2-7B | |
|---|---|---|---|---|---|
| | | PPL($\downarrow$) | Acc$_7$($\uparrow$) | PPL($\downarrow$) | Acc$_7$($\uparrow$) |
| FP16 | - | 6.13 | 70.05 | 5.47 | 66.69 |
| W4A16 | RTN | 8.52 | 65.32 | 5.73 | 62.54 |
| W4A16 | GPTQ | 6.54 | 68.72 | 5.69 | 64.89 |
| W4A16 | AWQ | 6.48 | 68.95 | 5.60 | 65.12 |
| W4A16 | OmniQuant | 6.50 | 68.60 | 5.58 | 64.94 |
| LaRoSA$_{25\%}$ | - | 6.13 | 69.54 | 5.51 | 66.39 |
| LaRoSA$_{25\%}$ | RTN | 8.64 | 64.90 | 5.81 | 62.19 |
| LaRoSA$_{25\%}$ | GPTQ | 6.58 | 68.20 | 5.74 | 64.64 |
| LaRoSA$_{25\%}$ | AWQ | 6.56 | 68.44 | 5.70 | 64.90 |
| LaRoSA$_{25\%}$ | OmniQuant | 6.60 | 68.32 | 5.68 | 64.86 |
| LaRoSA$_{50\%}$ | - | 7.22 | 67.19 | 5.87 | 64.61 |
| LaRoSA$_{50\%}$ | RTN | 9.54 | 62.40 | 6.10 | 60.43 |
| LaRoSA$_{50\%}$ | GPTQ | 7.64 | 65.87 | 6.07 | 62.91 |
| LaRoSA$_{50\%}$ | AWQ | 7.58 | 66.10 | 6.01 | 63.80 |
| LaRoSA$_{50\%}$ | OmniQuant | 7.62 | 66.14 | 5.96 | 63.52 |

formance. This ablation study highlights the robustness of our LaRoSA approach.

**Compatiblity with Quantization**. We demonstrate that LaRoSA also shows high compatibility with latest SOTA weight quantization methods. In detail, we experimented with LaRoSA using 4-bit quantization(W4A16) on model weights by employing RTN (Dettmers et al., 2022), GPTQ (Frantar et al., 2022), AWQ (Lin et al., 2024b) and OmniQuant (Shao et al., 2024) methods. Table 9 displays the perplexity and zero-shot accuracy results. With 4-bit GPTQ quantization, LaRoSA achieves 64.64% zero-shot accuracy and 5.74 PPL for LLaMA2-7B at 25% sparsity. These results are nearly on par with the the GPTQ quantized model's 64.89% accuracy and 5.69 PPL. This indicates that the empirical errors from sparsity and quantization are somewhat orthogonal to each other. The compatibility of LaRoSA with weight quantization highlights potential efficiency improvements through optimized kernels that integrate both sparsification and quantization operations.

## 6. Conclusion

This work presents LaRoSA, a training-free activation sparsification method that significantly enhances the efficiency of LLM inference. By utilizing orthogonal rotation matrices and Top-K sparsification functions, LaRoSA effectively optimizes sparse patterns for input activations, achieving minimal performance degradation along with consistent inference acceleration. Extensive experiments demonstrate that LaRoSA is robust and effective across various model types, sizes, and sparsity levels, offering superior performance and notable improvements in wall-clock time speedup. LaRoSA establishes new state-of-the-art results in activation sparsification scenarios, enhancing the deployment of efficient LLMs in resource-constrained environments.

## Impact Statement

This paper aims to contribute to the advancement of Machine Learning through the exploration of activation sparsification techniques. These techniques have the potential to enhance model efficiency and scalability, leading to broader applicability across various domains. While there are numerous possible societal implications of our work, we do not identify any that require specific emphasis at this time.

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

## A. Theoretical Analysis

This section provides a detailed explanation of how LaRoSA utilizes orthogonal transformations and top-k sparsification, and why it achieves consistently lower empirical errors compared to TEAL (Liu et al., 2024a).

TEAL derives the theoretical errors from magnitude-based sparsification under a restrictive assumption that weights and activations are independent and identically Gaussian distributed. However, we found that empirical errors are generally larger than theoretical errors, as shown in Figure 5, which may be due to the incorrect assumption of a Gaussian distribution.

In fact, the rotated activation through orthogonal transformations follows an approximately Gaussian distribution. Specifically, elements of $\tilde{x} = xQ$ are the weighted averages of elements of $x$. That is, for any $i$, $\tilde{x}_i = \sum_{j=1}^{D_{in}} x_j Q_{ji}$, where $\sum_{j=1}^{D_{in}} Q_{ji}^2 = 1$. By the Central Limit Theorem, if $x_i$ is independent and identically distributed (i.i.d.) random variables with zero mean, then $\tilde{x}_i$ will follow an approximately zero-mean Gaussian distribution.

After rotation, only top $k$ elements of $\tilde{x}$ are retained. It can be shown that under the assumption that $\tilde{x}$ and $\tilde{W} = WQ$ are are independent and identically Gaussian distributed, the theoretical error of top $k$ sparsification is determined solely by $k$.

**Theorem A.1.** *Let $\tilde{x} \in \mathbb{R}^{D_{in}}$, $\tilde{W} \in \mathbb{R}^{D_{in} \times D_{out}}$ are independent and identically Gaussian distributed, where $\tilde{x}_i \sim N(0, \sigma_x)$ and $\tilde{W}_{ji} \sim N(0, \sigma_w)$. Furthermore, for top-k sparsification method, we assume $\mathbb{E}[\tilde{x}_i^2|\tilde{x}_i \notin Top_k(\tilde{x})] = \mathbb{E}[\tilde{x}_i^2||\tilde{x}_i| < t_k]$, where $t_k$ satisfies $\mathbb{P}(|\tilde{x}_i| \geq t_k) = \frac{k}{D_{in}}$. For sparsification function $S_k$, define $y^S = S_k(\tilde{x})\tilde{W}^T$, and $y = \tilde{x}\tilde{W}^T$, then the distributional relative error is given by:*

$$\frac{\mathbb{E}\|y - y^S\|_2}{\mathbb{E}\|y\|_2} = \sqrt{1 - \frac{k}{D_{in}} - 2\Phi^{-1}\left(1 - \frac{k}{2D_{in}}\right)\varphi\left(\Phi^{-1}\left(1 - \frac{k}{2D_{in}}\right)\right)} \tag{11}$$

*where $\varphi(t) = \frac{1}{\sqrt{2\pi}}e^{-\frac{1}{2}t^2}$ is the standard Gaussian probability density function, and $\Phi^{-1}$ is the inverse cumulative distribution function (CDF) of the standard Gaussian distribution.*

*Proof.* define $\hat{y} = y - y^S$. Due to independence and zero mean of $\tilde{W}$, for any $j \neq k$ and any $i$, it follows that $\mathbb{E}[\tilde{W}_{ji}\tilde{W}_{ki}] = 0$, which leads to:

$$\mathbb{E}[\hat{y}_j\hat{y}_k] = \mathbb{E}\left[\sum_{i_1=1}^{D_{in}}(\tilde{x}_{i_1} - S_k(\tilde{x}_{i_1}))\tilde{W}_{ji_1}\sum_{i_2=1}^{D_{in}}(\tilde{x}_{i_2} - S_k(\tilde{x}_{i_2}))\tilde{W}_{ki_2}\right]$$

$$= \mathbb{E}\left[\sum_{i_1=1}^{D_{in}}\sum_{i_2=1}^{D_{in}}(\tilde{x}_{i_1} - S_k(\tilde{x}_{i_1}))(\tilde{x}_{i_2} - S_k(\tilde{x}_{i_2}))\tilde{W}_{ji_1}\tilde{W}_{ki_2}\right]$$

$$= 0 \tag{12}$$

Also for any $i$, according to Lemma A.1 in (Liu et al., 2024a), we have :

$$\mathbb{E}[\hat{y}_i^2] = \mathbb{E}\left[\sum_{j=1}^{D_{in}}\left((\tilde{x}_j - S_k(\tilde{x}_j))\tilde{W}_{ji}\right)^2\right]$$

$$= \sigma_w^2\sum_{j=1}^{D_{in}}\mathbb{E}\left[\tilde{x}_j - S_k(\tilde{x}_j)\right]^2$$

$$= \sigma_w^2\sum_{j=1}^{D_{in}}\mathbb{E}[\tilde{x}_j^2|\tilde{x}_j \notin Top_k(\tilde{x})]$$

$$= \sigma_w^2\sum_{j=1}^{D_{in}}\mathbb{E}\left[\tilde{x}_j^2||\tilde{x}_j| < t_k\right]$$

$$= D_{in}\sigma_x^2\sigma_w^2\left(1 - \frac{k}{D_{in}} - \frac{2t_k}{\sigma_x}\varphi\left(\frac{t_k}{\sigma_x}\right)\right) \tag{13}$$

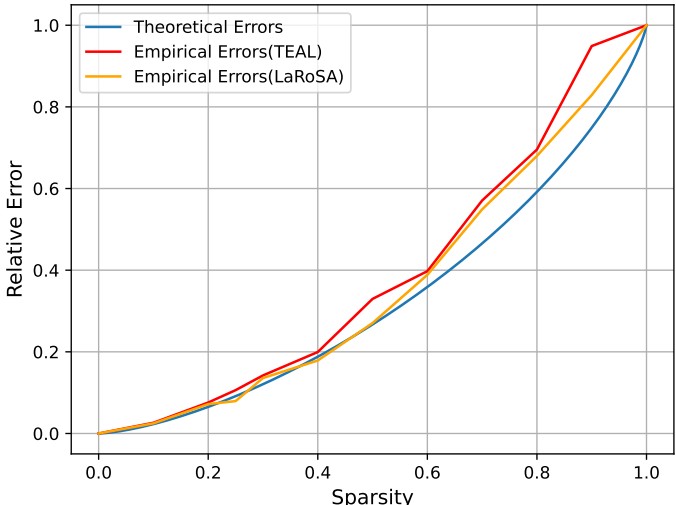

*Figure 5.* Relative output error results from 14th Layer of LLaMA3-8B. Theoretical errors are calculated with Top-K based sparsification. Empirical errors are derived from the output of $\mathbf{W}_{\text{down}}$ projection.

On the other hand, since $\tilde{\boldsymbol{x}}_i$ are Gaussian distributed, $1 - \frac{k}{D_{\text{in}}} = \mathbb{P}(|\tilde{\boldsymbol{x}}_i| \leq t_k) = \mathbb{P}(|z| \leq \frac{t_k}{\sigma_x}) = 2\Phi(\frac{t_k}{\sigma_x}) - 1$, where $z$ is a standard Gaussian random variable. It follows that:

$$\frac{t_k}{\sigma_x} = \Phi^{-1}\left(1 - \frac{k}{2D_{\text{in}}}\right) \tag{14}$$

As a result, we have:

$$\mathbb{E}\|\hat{\boldsymbol{y}}\|^2 = \sum_{i=1}^{D_{\text{out}}} \mathbb{E}[\hat{\boldsymbol{y}}_i]^2$$

$$= D_{\text{in}}D_{\text{out}}\sigma_x^2\sigma_w^2\left(1 - \frac{k}{D_{\text{in}}} - 2\Phi^{-1}\left(1 - \frac{k}{2D_{\text{in}}}\right)\varphi\left(\Phi^{-1}\left(1 - \frac{k}{2D_{\text{in}}}\right)\right)\right) \tag{15}$$

Note that

$$\mathbb{E}\|\boldsymbol{y}\|^2 = D_{\text{in}}D_{\text{out}}\sigma_x^2\sigma_w^2 \tag{16}$$

as the unsparsified case. Equation (15) and Equation (16) lead to the final result. $\qquad\square$

As illustrated in Figure 5, the improved Gaussian distribution assumption results in LaRoSA having consistent lower empirical errors compared to TEAL, bringing our proposed method closer to the theoretical errors. We employ magnitude-based top-K method instead of pruning the last few elements after rotation like SliceGPT (Ashkboos et al., 2024a), which ensures the validity of the aforementioned theoretical error analysis. Our experiments across multiple models demonstrate that, after rotating into the principal component space, most small-magnitude elements of activation are indeed concentrated in the last few dimensions.

## B. Grid Search Optimized Top-K

In the work of TEAL, it was noted that even under a target constraint of 50% layer sparsity, the sparsity of the four input activations (i.e., $h_1, h_2, h_3$, and $h_4$) still exhibit differing numerical distributions. Our exploration of the TopK approach, based on LaRoSA's framework, also revealed that forcing these four input activations to adopt the same TopK value can lead to suboptimal sparsity patterns in some models, even when corrective rotation matrices are introduced. To address this issue, we conducted a straightforward grid search for the sparsity coefficients $\alpha_1, \alpha_2, \alpha_3$, and $\alpha_4$ that determines the allocation weights for $h_1, h_2, h_3$, and $h_4$ across all layers. To ensure that the sparsity within the block aligns with the model-level sparsity, we need to consider the proportional relationship between the dimension size of $h_1$ and $h_2$ (as well as $h_3$ and $h_4$). Specifically, for input activations of Q, K, V, and O projections, we have:

$$3 * \alpha_1 + \alpha_2 = 4. \tag{17}$$

For input activations in Up, Gate, and Down projections, we have

$$2 * \alpha_3 + M * \alpha_4 = 2 + M, \tag{18}$$

where $M$ represents the ratio of the intermediate size to the hidden size. Thus, given $\alpha_1$ and $\alpha_3$, we have:

$$\alpha_2 = 4 - 3 * \alpha_1, \tag{19}$$

$$\alpha_4 = (2 + M - 2 * \alpha_3)/M. \tag{20}$$

After obtaining the optimal coefficients $\alpha$ for $h_1, h_2, h_3$, and $h_4$, we apply the same allocation weights across all layers. The experimental results in Table 10 demonstrated that our sparsity coefficients search strategy effectively improves the performance of the sparsified model. In Table 11, we provide a detailed presentation of the sparsity coefficients used across various models.

*Table 10.* LLaMA3-8B perplexity results on different sets of coefficients $\alpha$ for $h_1$, $h_2$, $h_3$, and $h_4$

| Method | Attention | | MLP | | PPL |
|---|---|---|---|---|---|
| | $h_1$ | $h_2$ | $h_3$ | $h_4$ | |
| TopK$_{60\%}$ | 1.00 | 1.00 | 1.00 | 1.00 | 9.09 |
| TopK+GS$_{60\%}$ | 0.90 | 1.30 | 0.90 | 1.06 | 8.98 |
| TopK+GS$_{60\%}$ | 0.85 | 1.45 | 0.70 | 1.17 | 8.94 |
| TopK+GS$_{60\%}$ | 0.80 | 1.60 | 0.80 | 1.12 | 8.86 |
| TopK+GS$_{60\%}$ | 0.75 | 1.75 | 0.75 | 1.14 | 8.93 |
| LaRoSA$_{60\%}$ | 0.80 | 1.60 | 0.80 | 1.12 | **8.57** |

*Table 11.* Optimal sparsity coefficients $\alpha$ of $h_1$, $h_2$, $h_3$, and $h_4$ used in our proposed LaRoSA.

| Models | Attention | | MLP | | |
|---|---|---|---|---|---|
| | $h_1$ | $h_2$ | $h_3$ | $h_4$ | M |
| LLaMA2-7B | 0.90 | 1.30 | 0.80 | 1.15 | 2.6875 |
| LLaMA2-70B | 0.80 | 1.60 | 0.80 | 1.12 | 3.5000 |
| LLaMA3-8B | 0.80 | 1.60 | 0.80 | 1.12 | 3.5000 |
| LLaMA3-70B | 0.85 | 1.45 | 0.75 | 1.14 | 3.5000 |
| Mistral-7B | 1.00 | 1.00 | 0.80 | 1.12 | 3.5000 |
| Qwen2.5-7B | 0.80 | 1.60 | 0.80 | 1.08 | 5.2857 |
| Qwen2.5-72B | 0.80 | 1.60 | 0.80 | 1.11 | 5.2857 |

---

**Algorithm 1** Grid Search for Optimal Sparsity Coefficients

---

1: **Input:** Initial Top-K function value $K$, Ratio of the intermediate size to the hidden size $M$, Step size $s = 0.05$
2: **Output:** Optimal sparsity coefficients $\alpha_1^*, \alpha_2^*, \alpha_3^*, \alpha_4^*$
3: Initialize best_PPL: $best\_ppl \leftarrow +\infty$
4: Initial sparsity coefficients $\alpha_1 = 0.7$, $\alpha_3 = 0.7$
5: **while** $\alpha_1 \leq 1.2$ **do**
6:     $\alpha_2 \leftarrow$ 4 - 3 * $\alpha_1$
7:     **while** $\alpha_3 \leq 1.2$ **do**
8:         $\alpha_4 \leftarrow$ (2 + M - 2 * $\alpha_3$)/$M$
9:         Compute current_PPL with new Top-K function value $\{\alpha_1 K, \alpha_2 K, \alpha_3 K, \alpha_4 K\}$ for $\boldsymbol{h}_1, \boldsymbol{h}_2, \boldsymbol{h}_3, \boldsymbol{h}_4$
10:        **if** current_PPL $<$ best_PPL **then**
11:            best_PPL $\leftarrow$ current_PPL
12:            $\alpha_1^* \leftarrow \alpha_1$
13:            $\alpha_3^* \leftarrow \alpha_3$
14:        **end if**
15:        $\alpha_3 \leftarrow \alpha_3 + s$
16:    **end while**
17:    $\alpha_1 \leftarrow \alpha_1 + s$
18: **end while**
19: $\alpha_2^* \leftarrow$ 4 - 3 * $\alpha_1^*$
20: $\alpha_4^* \leftarrow$ (2 + M - 2 * $\alpha_3^*$)/$M$
21: **Return** $\alpha_1^*, \alpha_2^*, \alpha_3^*, \alpha_4^*$

---

## C. Full Experimental Results

*Table 12.* Full results of LaRoSA in zero-shot and few-shot tasks for LLaMA2-7B.

| Sparsity | Method | WinoGrande | PiQA | OBQA | HellaSwag | BoolQ | ARC-E | ARC-C | **Avg** | MMLU |
|---|---|---|---|---|---|---|---|---|---|---|
| | Metrics | acc | acc_norm | acc_norm | acc_norm | acc | acc_norm | acc_norm | | acc |
| | Baseline | 69.14 | 79.05 | 44.20 | 75.98 | 77.71 | 74.54 | 46.25 | 66.69 | 45.85 |
| 25% | CATS | 67.64 | 77.80 | 41.20 | 75.75 | 70.55 | 69.70 | 43.34 | 63.71 | 42.76 |
| | TEAL | 67.95 | 78.18 | 44.00 | 75.20 | 76.71 | 73.91 | 45.64 | 65.94 | 44.66 |
| | LaRoSA | 68.43 | 79.05 | 44.00 | 75.84 | 77.22 | 74.66 | 45.56 | 66.39 | 45.66 |
| 40% | CATS | 56.12 | 66.87 | 32.80 | 52.28 | 63.21 | 44.32 | 31.23 | 49.55 | 24.67 |
| | TEAL | 66.93 | 77.96 | 42.60 | 74.09 | 75.90 | 73.31 | 43.68 | 64.92 | 43.46 |
| | LaRoSA | 68.90 | 78.18 | 44.20 | 75.35 | 76.85 | 73.86 | 45.73 | 66.15 | 44.66 |
| 50% | TEAL | 65.27 | 77.53 | 41.20 | 71.39 | 73.33 | 71.17 | 42.66 | 63.22 | 39.57 |
| | LaRoSA | 67.32 | 77.69 | 43.40 | 74.37 | 73.85 | 72.31 | 43.34 | 64.61 | 43.10 |

*Table 13.* Full results of LaRoSA in zero-shot and few-shot tasks for LLaMA2-70B.

| Sparsity | Method | WinoGrande | PiQA | OBQA | HellaSwag | BoolQ | ARC-E | ARC-C | Avg | MMLU |
|---|---|---|---|---|---|---|---|---|---|---|
| | | acc | acc_norm | acc_norm | acc_norm | acc | acc_norm | acc_norm | | acc |
| | Baseline | 77.97 | 82.80 | 48.80 | 83.81 | 83.79 | 81.10 | 57.33 | 73.66 | 68.80 |
| 25% | CATS | 76.32 | 82.26 | 49.20 | 84.22 | 79.94 | 79.63 | 57.59 | 72.74 | 67.50 |
| | TEAL | 78.56 | 82.42 | 48.80 | 82.66 | 83.25 | 81.26 | 56.24 | 73.31 | 67.90 |
| | LaRoSA | 57.17 | 80.09 | 83.88 | 83.63 | 48.60 | 82.70 | 77.58 | 73.38 | 68.74 |
| 40% | CATS | 67.25 | 77.69 | 43.00 | 75.11 | 71.41 | 61.11 | 43.60 | 62.74 | 55.83 |
| | TEAL | 76.21 | 81.65 | 48.30 | 82.02 | 82.94 | 80.87 | 55.30 | 72.47 | 66.78 |
| | LaRoSA | 77.43 | 83.57 | 47.40 | 83.37 | 83.36 | 80.51 | 57.76 | 73.31 | 68.16 |
| 50% | TEAL | 75.74 | 81.33 | 47.40 | 80.67 | 81.80 | 80.35 | 56.16 | 71.92 | 64.43 |
| | LaRoSA | 76.80 | 82.86 | 48.80 | 82.99 | 82.78 | 79.63 | 56.14 | 72.86 | 67.57 |

*Table 14.* Full results of LaRoSA in zero-shot and few-shot tasks for LLaMA3-8B.

| Sparsity | Method | WinoGrande | PiQA | OBQA | HellaSwag | BoolQ | ARC-E | ARC-C | Avg | MMLU |
|---|---|---|---|---|---|---|---|---|---|---|
| | | acc | acc_norm | acc_norm | acc_norm | acc | acc_norm | acc_norm | | acc |
| | Baseline | 73.64 | 80.41 | 44.80 | 79.13 | 81.10 | 77.78 | 53.50 | 70.05 | 65.26 |
| 25% | CATS | 70.24 | 79.33 | 44.40 | 77.38 | 79.17 | 73.70 | 49.40 | 67.66 | 61.85 |
| | TEAL | 72.48 | 80.36 | 45.40 | 78.45 | 80.07 | 77.08 | 52.00 | 69.40 | 63.85 |
| | LaRoSA | 72.93 | 80.63 | 45.00 | 79.09 | 80.40 | 76.68 | 52.05 | 69.54 | 64.85 |
| 40% | CATS | 60.54 | 73.12 | 36.00 | 66.82 | 59.14 | 52.78 | 37.37 | 55.11 | 31.82 |
| | TEAL | 70.34 | 78.51 | 45.80 | 76.55 | 79.57 | 76.47 | 49.74 | 68.14 | 59.84 |
| | LaRoSA | 71.11 | 79.87 | 44.00 | 77.85 | 80.09 | 76.98 | 51.62 | 68.79 | 62.61 |
| 50% | TEAL | 66.71 | 77.09 | 42.80 | 72.61 | 76.26 | 72.71 | 46.24 | 64.92 | 52.78 |
| | LaRoSA | 68.98 | 79.98 | 44.20 | 76.85 | 77.55 | 74.75 | 48.04 | 67.19 | 58.65 |

*Table 15.* Full results of LaRoSA in zero-shot and few-shot tasks for LLaMA3-70B.

| Sparsity | Method | WinoGrande acc | PiQA acc_norm | OBQA acc_norm | HellaSwag acc_norm | BoolQ acc | ARC-E acc_norm | ARC-C acc_norm | Avg | MMLU acc |
|---|---|---|---|---|---|---|---|---|---|---|
| | Baseline | 80.11 | 84.49 | 48.80 | 84.97 | 85.22 | 86.15 | 64.33 | 76.29 | 78.71 |
| 25% | CATS | 79.87 | 84.06 | 47.20 | 85.21 | 84.43 | 85.19 | 63.14 | 75.58 | 77.93 |
| | TEAL | 76.63 | 82.31 | 48.2 | 81.82 | 83.24 | 81.64 | 58.78 | 73.23 | 74.86 |
| | LaRoSA | 81.45 | 84.39 | 49.00 | 85.07 | 84.89 | 85.65 | 63.65 | 76.30 | 78.13 |
| 40% | CATS | 74.51 | 81.77 | 46.40 | 83.46 | 81.44 | 79.17 | 57.76 | 72.07 | 72.12 |
| | TEAL | 76.32 | 81.93 | 45.80 | 81.56 | 80.64 | 81.90 | 57.50 | 72.24 | 73.23 |
| | LaRoSA | 79.72 | 83.62 | 48.40 | 84.71 | 85.05 | 84.26 | 62.12 | 75.41 | 77.62 |
| 50% | TEAL | 73.08 | 80.30 | 43.60 | 79.90 | 81.74 | 80.17 | 56.82 | 70.80 | 69.20 |
| | LaRoSA | 77.90 | 82.59 | 46.40 | 84.24 | 84.10 | 82.49 | 58.96 | 73.81 | 76.51 |

*Table 16.* Full results of LaRoSA in zero-shot and few-shot tasks for Mistral-7B-v0.3.

| Sparsity | Method | WinoGrande acc | PiQA acc_norm | OBQA acc_norm | HellaSwag acc_norm | BoolQ acc | ARC-E acc_norm | ARC-C acc_norm | Avg | MMLU acc |
|---|---|---|---|---|---|---|---|---|---|---|
| | Baseline | 73.80 | 82.31 | 44.00 | 80.40 | 82.11 | 78.40 | 52.04 | 70.44 | 62.34 |
| 25% | CATS | 71.27 | 80.47 | 33.00 | 60.68 | 79.42 | 78.28 | 48.72 | 64.55 | 59.83 |
| | TEAL | 72.61 | 81.82 | 44.40 | 80.05 | 81.44 | 78.24 | 51.86 | 70.06 | 61.51 |
| | LaRoSA | 72.85 | 82.15 | 44.40 | 80.14 | 81.71 | 78.49 | 52.05 | 70.25 | 61.81 |
| 40% | CATS | 61.96 | 74.43 | 34.80 | 74.71 | 74.86 | 55.13 | 41.47 | 59.62 | 44.31 |
| | TEAL | 69.55 | 80.36 | 44.40 | 79.08 | 80.49 | 77.14 | 50.30 | 68.76 | 60.17 |
| | LaRoSA | 70.88 | 81.07 | 43.40 | 79.63 | 81.68 | 78.32 | 51.11 | 69.44 | 61.15 |
| 50% | TEAL | 68.71 | 79.52 | 40.40 | 76.97 | 79.55 | 74.34 | 47.63 | 66.73 | 57.34 |
| | LaRoSA | 69.93 | 81.61 | 43.20 | 78.93 | 80.92 | 75.55 | 49.06 | 68.46 | 58.80 |

*Table 17.* Full results of LaRoSA in zero-shot and few-shot tasks for Qwen2.5-7B.

| Sparsity | Method | WinoGrande acc | PiQA acc_norm | OBQA acc_norm | HellaSwag acc_norm | BoolQ acc | ARC-E acc_norm | ARC-C acc_norm | Avg | MMLU acc |
|---|---|---|---|---|---|---|---|---|---|---|
| | Baseline | 72.77 | 79.98 | 47.60 | 78.91 | 84.56 | 77.44 | 51.11 | 70.34 | 74.21 |
| 25% | CATS | 70.64 | 79.54 | 46.20 | 78.90 | 83.15 | 78.28 | 50.94 | 69.66 | 72.67 |
| | TEAL | 72.12 | 79.22 | 46.40 | 78.49 | 84.40 | 76.85 | 50.85 | 69.76 | 73.21 |
| | LaRoSA | 72.69 | 79.92 | 46.60 | 78.77 | 84.40 | 77.02 | 51.45 | 70.12 | 73.74 |
| 40% | CATS | 58.72 | 76.12 | 38.60 | 72.49 | 74.01 | 67.89 | 44.97 | 61.83 | 63.99 |
| | TEAL | 70.30 | 78.78 | 45.00 | 76.78 | 84.05 | 75.44 | 49.96 | 68.61 | 71.44 |
| | LaRoSA | 70.72 | 79.38 | 46.20 | 78.28 | 84.65 | 77.86 | 50.60 | 69.67 | 72.33 |
| 50% | TEAL | 68.90 | 77.18 | 44.00 | 74.60 | 83.76 | 76.59 | 49.34 | 67.76 | 68.53 |
| | LaRoSA | 70.24 | 79.11 | 45.60 | 77.20 | 83.55 | 78.03 | 49.91 | 69.09 | 70.09 |

*Table 18.* Full results of LaRoSA in zero-shot and few-shot tasks for Qwen2.5-72B.

| Sparsity | Method | WinoGrande acc | PiQA acc_norm | OBQA acc_norm | HellaSwag acc_norm | BoolQ acc | ARC-E acc_norm | ARC-C acc_norm | Avg | MMLU acc |
|---|---|---|---|---|---|---|---|---|---|---|
| | Baseline | 77.90 | 83.62 | 46.40 | 86.05 | 89.14 | 83.33 | 62.62 | 75.58 | 86.08 |
| 25% | CATS | 73.72 | 82.86 | 45.40 | 86.60 | 82.57 | 79.92 | 58.36 | 72.77 | 84.91 |
| | TEAL | 78.11 | 83.39 | 46.50 | 85.23 | 87.63 | 82.62 | 61.87 | 75.05 | 85.44 |
| | LaRoSA | 77.66 | 83.73 | 46.80 | 85.97 | 89.17 | 82.83 | 62.54 | 75.53 | 85.62 |
| 40% | CATS | 65.59 | 78.56 | 33.20 | 61.81 | 82.14 | 74.71 | 45.82 | 63.12 | 80.95 |
| | TEAL | 77.47 | 83.01 | 46.50 | 84.52 | 87.52 | 82.43 | 61.11 | 74.65 | 84.80 |
| | LaRoSA | 78.45 | 83.03 | 47.20 | 86.01 | 89.48 | 83.12 | 62.46 | 75.35 | 85.33 |
| 50% | TEAL | 77.32 | 82.25 | 45.00 | 83.03 | 87.54 | 81.61 | 59.46 | 73.74 | 83.54 |
| | LaRoSA | 76.48 | 83.57 | 45.60 | 85.81 | 89.51 | 83.59 | 61.69 | 75.18 | 84.34 |

*Table 19.* The change in TFLOPs of the model after adding the residual adaptor.

| Model | Prefill | | Decode | |
|---|---|---|---|---|
| | Original (TFLOPs) | LaRoSA (TFLOPs) | Original (TFLOPs) | LaRoSA (TFLOPs) |
| LLaMA2-7B | 29.26(1.00×) | 31.46(1.07×) | 29.26(1.00×) | 31.46(1.07×) |
| LLaMA2-70B | 292.44(1.00×) | 314.43(1.07×) | 292.45(1.00×) | 314.44(1.07×) |
| LLaMA3-8B | 32.94(1.00×) | 35.14(1.06×) | 32.94(1.00×) | 35.14(1.06×) |
| LLaMA3-70B | 295.67(1.00×) | 317.66(1.07×) | 295.68(1.00×) | 317.67(1.07×) |
| Mistral-7B | 31.34(1.00×) | 33.53(1.07×) | 31.34(1.00×) | 33.54(1.07×) |
| Qwen2.5-7B | 30.64(1.00×) | 32.11(1.05×) | 30.64(1.00×) | 32.12(1.05×) |
| Qwen2.5-72B | 303.69(1.00×) | 32.57(1.07×) | 303.69(1.00×) | 325.68(1.07×) |

