# OpenReview forum: "La RoSA: Enhancing LLM Efficiency via Layerwise Rotated Sparse Activation"
_ICML.cc/2025/Conference — ICML 2025 poster_

### Official Review · Reviewer_jWLg · 2025-03-12

**Overall Recommendation:** 4

**Summary:**

The paper introduces LaRoSA, a method that applies layer-wise rotation to activation sparsity, which improves compression performance with minimal additional inference cost.

**Claims And Evidence:**

The claims are well-supported with clear evidence.

**Essential References Not Discussed:**

It might be beneficial to include a discussion on pruning methods in related works, which could further contextualize the contributions of LaRoSA.

**Experimental Designs Or Analyses:**

The experimental results for TEAL are slightly worse than those reported in the original paper. Given the reproducibility concerns, I suggest that the authors provide a discussion and analysis regarding the potential reasons for this discrepancy.

**Methods And Evaluation Criteria:**

The evaluations include different series of LLMs, demonstrating the effectiveness of LaRoSA across various architectures.

**Other Comments Or Suggestions:**

Please see the questions below for additional feedback.

**Other Strengths And Weaknesses:**

The paper is well-structured and clearly written, making it easy to follow and understand.

**Questions For Authors:**

1. Could LaRoSA be combined with semi-structured pruning methods, such as Wanda [1] and RIA [2]? If possible, could you provide experimental results to support this?
2. I am somewhat confused about Figure 3. Could you provide more analysis of the experimental details and insights from the data presented?
3. The GPTQ baseline in Table 7 is not state-of-the-art. It would be valuable to explore and compare with more advanced methods.


[1] A Simple and Effective Pruning Approach for Large Language Models, ICLR 2024.

[2] Plug-and-Play: An Efficient Post-training Pruning Method for Large Language Models, ICLR 2024.

**Relation To Broader Scientific Literature:**

This paper follows the main pipeline of CATS and TEAL, aiming to improve activation sparsity.

**Theoretical Claims:**

I have reviewed the proof section in the Appendix.

---

> ### Author Rebuttal · Authors · 2025-03-28
>
> We sincerely thank reviewer jWLg for the thorough review! Below is our point-by-point response to your feedback:
>
> > The experimental results for TEAL are slightly worse than those reported in the original paper ...
>
> Yes, your observation is correct. We will modify and extend implementation details in future revisions.
>
> As noted in Section 5.4.3 of the TEAL paper, TEAL applies activation sparsification to only 99% of tokens during the prefill stage. The authors attribute this to attention sinks[3], which cause *'more severe degradation when sparsifying the initial tokens'*. Therefore, they skipped the first 1% of tokens to maintain model performance. In Section 4.1 of the LaRoSA paper, we argue that this is due to **a mismatch between the actual and calibrated thresholds for each token.** As shown in Figure 3b, this mismatch is especially pronounced at the start of sequences, leading to TEAL's poor performance on those initial tokens.
>
> In LaRoSA and CATS, **all tokens are sparsified without special treatment**. For a fair comparison, we applied full activation sparsification for TEAL, resulting in slightly worse results than TEAL's report. This also reflects the lack of robustness in using empirical thresholds.
>
> > Could LaRoSA be combined with semi-structured pruning methods such as Wanda and RIA?
>
> Thank you for the kind suggestion. It is indeed possible. Combining LaRoSA with other model compression methods is a promising optimization direction. We conduct experiments using RIA[2] repository, and randomly choose 128 sequences from C4 dataset to compute importance matrices for Wanda and RIA. We applied LaRoSA after semi-structured pruning. The results are summarized as follows:
>
> |Sparsity|Method||LLaMA3|-8B||Qwen2.5|-7B|
> |-|-|-|-|-|-|-|-|
> |||PPL(↓)|Acc(↑)|Speed-Up(↑)|PPL(↓)|Acc(↑)|Speed-Up(↑)|
> |Dense|-|6.1|70.0|1.00×|6.8|70.3|1.00×|
> |LaRoSA 25%|-|6.2|69.5|1.14×|6.9|70.1|1.15×|
> |LaRoSA 40%|-|6.6|68.8|1.30×|7.1|69.6|1.32×|
> |2:4|Wanda|24.2|62.3|1.24×|25.6|59.3|1.25×|
> |+LaRoSA 25%|Wanda|24.5|61.8|1.33×|26.0|59.0|1.35×|
> |+LaRoSA 40%|Wanda|25.0|60.1|1.40×|26.5|57.4|1.42×|
> |2:4|RIA|23.0|63.1|1.23×|24.1|60.1|1.25×|
> |+LaRoSA 25%|RIA|23.2|62.9|1.34×|24.4|59.8|1.34×|
> |+LaRoSA 40%|RIA|23.4|61.6|1.40×|24.9|57.9|1.41×|
>
> Although Wanda and RIA achieve speed-up by using the cuTLASS library, these methods experience notable performance drop, which also aligns with recent findings reported in the RIA's repo for latest models.
>
> It's important to note that while Wanda and RIA focus on weight pruning, LaRoSA is an activation sparsification method. Due to different pruning targets, combining these methods does not lead to severe error accumulation. This highlight LaRoSA's compatibility with different compression methods. Moreover, at similar sparsity levels, LaRoSA's 40% achieved higher speed-up with less performance loss compared to RIA 2:4 (50%). This indicates: **1) The customized kernel in LaRoSA effectively addresses many inference bottlenecks in LLMs. 2) Activation sparsification could offer performance benefits over weight pruning, suggesting a promising direction for optimizing LLM inference.**
>
> > I am somewhat confused about Figure 3. Could you provide more analysis?
>
> We apologize for not providing the best interpretation of Figure 3 in the current submission. We have detailed the experimental setup, motivations, and insights in our response to reviewer V3DZ. We kindly ask you to refer to our response to reviewer V3DZ for more information.
>
> > The GPTQ baseline in Table 7 is not state-of-the-art.
>
> We appreciate the reviewer's constructive suggestions. We have added experiments using AWQ[4] and OmniQuant[5] for weight-only quantization. We will also include these results in future versions of the paper.
>
> |Precision & Sparsity|Methods|LLaMA3-8B PPL(↓)|LLaMA3-8B Acc(↑)|LLaMA2-7B PPL(↓)|LLaMA2-7B Acc(↑)|
> |--|--|--|--|--|--|
> |FP16|-|6.13|70.05|5.47|66.69|
> |W4A16|AWQ|6.48|68.95|5.60|65.12|
> |W4A16|OmniQuant|6.50|68.60|5.58|64.94|
> |+LaRoSA 25%|AWQ|6.56|68.44|5.70|64.90|
> |+LaRoSA 25%|OmniQuant|6.60|68.32|5.68|64.86|
> |+LaRoSA 50%|AWQ|7.58|66.10|6.01|63.80|
> |+LaRoSA 50%|OmniQuant|7.62|66.14|5.96|63.52|
>
> It is evident that LaRoSA is highly compatible with the latest SOTA quantization methods. This further indicates that the **empirical errors introduced by activation sparsification and weight quantization are orthogonal**, suggesting that future works should consider both quantization and activation sparsification to achieve more efficient and reliable model inference.
>
> [1] A Simple and Effective Pruning Approach for Large Language Models, ICLR 2024.
> [2] Plug-and-Play: An Efficient Post-training Pruning Method for Large Language Models, ICLR 2024.
> [3] Efficient Streaming Language Models with Attention Sinks, ICLR 2024.
> [4] AWQ: Activation-aware Weight Quantization for On-Device LLM Compression and Acceleration, MLSys 2024.
> [5] OmniQuant: Omnidirectionally Calibrated Quantization for Large Language Models, ICLR 2024.

---

### Official Review · Reviewer_Yw7t · 2025-03-12

**Overall Recommendation:** 4

**Summary:**

The paper presents LaRoSA, a training-free activation sparsification method aimed at improving inference efficiency in LLMs. The key idea is orthogonal rotation-based pruning using a Top-K selection mechanism that eliminates lower-magnitude activations. The experimental results across multiple LLMs (LLaMA2, LLaMA3, Mistral, Qwen2.5) demonstrate superior accuracy trade-off compared to existing approaches like CATS and TEAL​.

## update after rebuttal

The authors have answered my concerns, and I still believe this paper merits acceptance and is a good paper.

**Claims And Evidence:**

All claims are well-supported as far as I can tell.

**Essential References Not Discussed:**

There are many works in the field of pruning not discussed or compared to, but I acknowledge the authors cannot be expected to discuss every single one.

**Experimental Designs Or Analyses:**

The experimental designs seem solid and robust.
The authors test on four major LLM architectures (LLaMA2, LLaMA3, Mistral, Qwen2.5) of various sizes (7B-70B). They compare their method with the TEAL and CATS baseline methods under the same sparsity constraints.
Most importantly, they measure performance on a large set of downstream tasks.

**Methods And Evaluation Criteria:**

The methods and evaluation criteria in the paper make sense.
While I'm skeptic of PPL results alone, the authors included various benchmark results (ARC, MMLU, HellaSwag, BoolQ, etc.).
All results are compared to baseline methods.

**Other Comments Or Suggestions:**

Appendix A, minor: I'd include a 1-line at the top to explain what this appendix section discusses.

**Other Strengths And Weaknesses:**

Strengths:

-No retraining required.

-Outperforms baseline methods on a large set of benchmarks.

-Provides kernels to utilize the method, something not every paper may do.

-The authors include various ablations to show their method is compatible with quantization and the effects of various choices.

-I appreciate the theoretical analysis in appendix A.

Weaknesses:

-The authors correctly criticize other works for distributional shifts between their calibration set and test time. But the method constructs the rotation matrix based on data. I don't think it's necessarily as bad, especially since the authors have addressed it in Table 6 (minor typo: the paper refers to Table 7 instead) to show the impact of the calibration data is minimal.

**Questions For Authors:**

None.

**Relation To Broader Scientific Literature:**

The work builds upon pruning techniques , and incorporates ideas inspired from rotation-based quantization.

**Theoretical Claims:**

The paper derives error bounds for Top-K of rotated activations, and provide mathematical derivations in Appendix A to support it. As far as I can tell, it seems correct.

---

> ### Author Rebuttal · Authors · 2025-03-29
>
> We sincerely thank the reviewer Yw7t for taking your valuable time to evaluate our work. The acknowledgment of our contributions has greatly inspired our entire team. Below is our point-by-point response to your feedback:
>
>
> >Appendix A, minor: I'd include a 1-line at the top to explain what this appendix section discusses.
>
> Thank you for your kind suggestion. In the next revision of the paper, we will add an introductory sentence to Appendix A: *'This section provides a detailed explanation of how LaRoSA utilizes orthogonal transformations and top-k sparsification, and why it achieves consistently lower empirical errors compared to TEAL.*
>
> >The authors correctly criticize other works for distributional shifts between their calibration set and test time. But the method constructs the rotation matrix based on data. I don't think it's necessarily as bad, especially since the authors have addressed it in Table 6 (minor typo: the paper refers to Table 7 instead) to show the impact of the calibration data is minimal.
>
> While developing our method, we shared the same concerns as the reviewer, which led us to include an ablation study focused on the choice of calibration datasets. Our paper's experimental results in Table 6 show that the method's performance is not sensitive to the specific calibration data used for creating the layerwise rotation matrix $Q_l$. To further demonstrate LaRoSA's effectiveness and robustness, we promptly carried out additional experiments on two distilled reasoning models, DeepSeek-R1-Distill-Llama3-8B[1] and DeepSeek-R1-Distill-Qwen2.5-7B[1], using the LaRoSA framework. We employed the Alpaca[2] dataset, which is not related to complex reasoning tasks, to obtain rotation matrix $Q_l$. We evaluate LaRoSA on harder tasks with these advanced models using Hugging Face Open-R1 repo.
>
> |Model|MATH 500|GPQA Diamond|AIME 2024|
> |-|-|-|-|
> |**DeepSeek-R1-Distill-Llama3-8B**||||
> |DeepSeek-R1 Report|89.1|49.0|50.4|
> |Reproduce Result|87.6|45.9|40.0|
> |+LaRoSA 25%|85.0|44.9|40.0|
> |+LaRoSA 40%|80.6|43.1|36.7|
> |+LaRoSA 50%|78.8|41.2|33.3|
> |**DeepSeek-R1-Distill-Qwen2.5-7B**||||
> |DeepSeek-R1 Report|92.8|49.1|55.5|
> |Reproduce Result|93.4|43.9|50.0|
> |+LaRoSA 25%|91.0|43.5|46.7|
> |+LaRoSA 40%|88.5|41.0|43.3|
> |+LaRoSA 50%|85.1|40.2|43.3|
>
> As shown in this table, our experiments demonstrated that the LaRoSA method maintains its effectiveness even when using data that is not closely related to complex reasoning tasks for generating the rotation matrix. We hope this set of experiments further alleviates the reviewers' concerns about the impact of calibration data on the performance of the LaRoSA algorithm.
>
> Once again, we express our gratitude to the reviewer Yw7t for acknowledging our paper's merits. We will continue to refine our work in the forthcoming stages and plan to open-source our code and models.
>
> [1] Deepseek-r1: Incentivizing reasoning capability in llms via reinforcement learning. arXiv preprint arXiv:2501.12948.
> [2] Stanford alpaca: An instruction-following llama model. https://github.com/tatsu-lab/stanford_alpaca, 2023.

---

> > ### Comment · Reviewer_Yw7t · 2025-04-02
> >
> > I thank the reviewers for answering my concerns, and still believe the paper merits acceptance.

---

### Official Review · Reviewer_V3DZ · 2025-03-13

**Overall Recommendation:** 3

**Summary:**

Sparse activation is an active research field of LLMs. This paper demonstrates that using magnitude-based pruning to sparsity leads to inconsistent sparsity across tasks and layers. Stemming from their rich preliminary study, they propose La RoSA which leverages the rotation operation proposed in SliceGPT to address the above issues. While rotation operation is not novel, extending it to activation sparsification to address inconsistent sparsity is novel. Extensive experiments are done to verify the effectiveness of La RoSA.

**Claims And Evidence:**

The overall claims made in the submission are supported with good experiments, with some minor claims needing to be clarified.

1. While perplexity is a good indicator for pre-training, it is not convincing enough to report PPL for LLM compression. See https://arxiv.org/pdf/2407.04965. I would like to see if sparsifying activations will cause any performance drop on harder tasks with more advanced models like DeepSeek-R1-Distill-Qwen on Math500 and GPQA.

2. While I appreciate the rich information provided in Figure 3, I found that we need more description for Figure 3 for readers to understand it fully. Figure 3b is a bit too complicated. I can grasp the main idea but need some time to interpret the figure and the corresponding text descriptions in the paper.  How many calibration data are used in Figure 3?

3. Could authors explain why larger sequence length leads to slower inference in Figure 3d? What is the role of inconsistent sparsity in this figure? Can the authors elaborate more on why we need consistent sparsity across layers to obtain a speed up?

4. I found the authors' claim in Section 4 is a bit controversial. At the beginning of Section 4, the authors mentioned that using a calibration dataset is not precise enough, leading to inaccurate sparsity. However, in the later part, the calibration dataset is used to perform Layerwise Orthogonal Rotation. Will this also lead to mismatching and performance drop?

5. I am wondering if the inconsistent sparsity of magnitude pruning can be mitigated with a longer calibration dataset?

6. The inference speed-up in Figure 4 is interesting, demonstrating the benefits of LaRoSA.

7. Another question is how sparse activation brings real speedup? I assume different tokens/samples will activate different sets of neurons. Will this constrain the practical usage of the proposed approach?

**Essential References Not Discussed:**

no

**Experimental Designs Or Analyses:**

I appreciate the evaluation of multiple architectures and various sizes. It seems LaRoSA is more effective at high sparsity levels than TEAL. I would like to see if the benefits also hold for math reasoning tasks, not only ppl.

**Methods And Evaluation Criteria:**

Using PCA instead of magnitude pruning appears to be a more precise option for sparse activations, as we have seen its effectiveness on structured pruning in SlideGPT.

The evaluation of this paper is relatively too easy. I would expect to see if the benefits still hold for complex reasoning tasks, such as math reasoning.

**Other Comments Or Suggestions:**

N/A

**Other Strengths And Weaknesses:**

**Strengths**

1. Related work is well-written and provides a clear summary of previous works in this field.

2. Experimental results seem to be strong, especially at high sparsity levels.

3. Inference speed is reported.

4. Ablation study explains the contribution of different components.

Weaknesses:

1. While perplexity is a good indicator for pre-training, it is not convincing enough to report PPL for LLM compression. See https://arxiv.org/pdf/2407.04965. I would like to see if sparsifying activations will cause any performance drop on harder tasks with more advanced models like DeepSeek-R1-Distill-Qwen on Math500 and GPQA.

2. While I appreciate the rich information provided in Figure 3, I found that we need more description for Figure 3 for readers to understand it fully. Figure 3b is a bit too complicated. I can grasp the main idea but need some time to interpret the figure and the corresponding text descriptions in the paper.  How many calibration data are used in Figure 3?

3. Could authors explain why larger sequence length leads to slower inference in Figure 3d? What is the role of inconsistent sparsity in this figure? Can the authors elaborate more on why we need consistent sparsity across layers to obtain a speed up?

4. I found the authors' claim in Section 4 is a bit controversial. At the beginning of Section 4, the authors mentioned that using a calibration dataset is not precise enough, leading to inaccurate sparsity. However, in the later part, the calibration dataset is used to perform Layerwise Orthogonal Rotation. Will this also lead to mismatching and performance drop?

5. I am wondering if the inconsistent sparsity of magnitude pruning can be mitigated with a longer calibration dataset?

6. The inference speed-up in Figure 4 is interesting, demonstrating the benefits of LaRoSA.

7. Another question is how sparse activation brings real speedup? I assume different tokens/samples will activate different sets of neurons. How can we address this mismatch in GPU?

**Questions For Authors:**

Please see the above weaknesses.

**Relation To Broader Scientific Literature:**

The main idea of this paper comes from the SliceGPT paper. Sparse activation is an active research field and I believe the authors have provided a quite good literature review.

**Theoretical Claims:**

N/A

---

> ### Author Rebuttal · Authors · 2025-03-28
>
> We thank reviewer for the thorough review! Below is our point-by-point response:
>
> > I would like to see ... on harder tasks with more advanced models.
>
> We agree that perplexity alone is insufficient, and follow your suggestion to evaluate LaRoSA on harder tasks with advanced models using Hugging Face Open-R1 repo.
>
> |Model|MATH500|GPQA Diamond|AIME 2024|
> |-|-|-|-|
> |**DeepSeek-R1-Distill-Llama3-8B**||||
> |DS-R1 Report|89.1|49.0|50.4|
> |Reproduce|87.6|45.9|40.0|
> |+LaRoSA 25%|85.0|44.9|40.0|
> |+LaRoSA 40%|80.6|43.1|36.7|
> |+LaRoSA 50%|78.8|41.2|33.3|
> |**DeepSeek-R1-Distill-Qwen2.5-7B**||||
> |DS-R1 Report|92.8|49.1|55.5|
> |Reproduce|93.4|43.9|50.0|
> |+LaRoSA 25%|91.0|43.5|46.7|
> |+LaRoSA 40%|88.5|41.0|43.3|
> |+LaRoSA 50%|85.1|40.2|43.3|
>
> LaRoSA preserves the math problem-solving and reasoning skills of distilled models, proving its robustness. Due to structural similarities with base models, their inference speedup matches the paper's results. These findings will be added in future revisions.
>
> > I found that we need more description for ... How many data are used?
>
> Figure 3b compares cutoff thresholds for activation pruning using TEAL and TopK methods at various sparsity levels. TEAL uses a fixed threshold from offline calibration, shown as a dashed line, while TopK computes thresholds during testing, shown as scatter points. We used 16 sequences of 512 tokens for various sparsity settings.
>
> **Note 1**: Ideally, actual thresholds should align with empirical ones at their sparsity levels. However, scatter points rarely match the dashed lines, indicating potential over-pruning.
>
> **Insight 1**: Empirical thresholds often misalign with real token activation, deviating from the intention of magnitude pruning.
>
> **Note 2**: Actual thresholds for initial tokens are often close to 0, and much lower than empirical thresholds.
>
> **Insight 2**: Empirical thresholds hold no real significance for initial tokens since they can zero out all activation values, harming model performance.
>
> We will modify and extend this part upon acceptance.
>
> > Why larger sequence length leads to slower inference?
>
> As context length increases, the proportion of attention computation and inference time required for each decoded token also grows. However, LaRoSA's speedup mainly comes from reduced memory access, which remains constant. Thus, acceleration benefits decrease with longer context lengths but still remain significantly faster than dense inference.
>
> > What is the role of inconsistent sparsity in this figure?
>
> In Figure 3d, inconsistent sparsity arises from varied token distributions across different data sources, like Code v.s. Chat tokens. The figure illustrates that empirical thresholds can not achieve the target sparsity on test datasets from different domains.
>
> > Why we need consistent sparsity across layers to obtain a speed up?
>
> 1）Inconsistent sparsity makes it difficult to ensure the model meets target sparsity, complicating efficiency predictions. 2）It results in variable inference times for each token. If a model compression method leads to uncertain and variable inference time for each token, it cannot guarantee efficiency and stability.
>
> > I found author claim ... Will this lead to mismatch and performance drop?
>
> In LaRoSA, calibration data is only used for PCA to compute rotation matrices. We shows that choice of datasets does not affect performance in Table6. Moreover, we test on DS-R1-Distill-Qwen2.5-7B with various datasets and reveal two findings: **1) Rotation matrices $Q_l$ are similar across calibration datasets. 2) Changing calibration data does not lead to mismatching or performance loss in LaRoSA.**
> |Method|Cali.Data|Avg.Cos.Simi|0-Shot Acc|MATH 500|GPQA Diamond|
> |-|-|-|-|-|-|
> |LaRoSA 25%|Alpaca|1.0(self)|82.9|91.0|43.5|
> |LaRoSA 25%|WikiText2|0.998|83.0|91.0|43.7|
> |LaRoSA 25%|C4|0.997|82.9|91.2|43.5|
>
> > I am wondering if the inconsistent ... with a longer calibration dataset?
>
> We tested different lengths of calibration data for Qwen2.5-7B and found that larger datasets did not resolve inconsistent sparsity or improve zero-shot performance. However, the source of the calibration data affects empirical thresholds, causing TEAL's performance to vary across datasets.
> |Method|Cali.Data/Size|16*2048|64*2048|256*2048|
> |-|-|-|-|-|
> |TEAL 25%|WikiText2|67.8|67.9|67.8|
> |TEAL 25%|Alpaca|69.7|69.7|69.6|
>
>
> > How sparse activation brings real speedup?
>
> The columns in the weight matrix related to zero-valued activations become inactive weights. Since the decode phase of LLM inference is memory-bound, avoiding the loading of these weights can bring real speedup.
>
> > I assume different tokens will ... How can we address this mismatch in GPU?
>
> Indeed, different tokens activate different neurons. Our custom kernel dynamically selects and loads only required weight columns for each token into the GPU. Unloaded weights are set to zero, making them inactive in GEMM while keeping dimensions unchanged. This prevents any potential mismatch in GPU.

---

### Official Review · Reviewer_nBNG · 2025-03-14

**Overall Recommendation:** 4

**Summary:**

This work proposes a sparse activation method for efficient inference by rotating inputs by an orthogonal matrix. Basic idea is to improve sparsity by rotating inputs to matrix multiplication happening in Transformer layers, e.g., MLP and attention, so that low magnitude inputs are pruned away in the rotated space without sacrificing representation in the original space. Experiments are carried out on diverse models and benchmarks demonstrating efficient inference without largely degrading performance when compared with other baselines.

**Claims And Evidence:**

The proposed method is inspired by the recent work on quantization in the rotated space (Ashkboos et al., 2024) and applied to sparse activation pruning of (Liu et al., 2025). The approach is apt and its effectiveness is supported by experiments.

**Essential References Not Discussed:**

No additional references are necessary.

**Experimental Designs Or Analyses:**

Experiments and analyses are exhaustive enough to demonstrate the effectiveness of the proposed approach.

**Methods And Evaluation Criteria:**

The approach is a sound method in that it is based on rotation by an orthogonal matrix estimated for each layer. Evaluation metrics are standard one to quantify the gains.

**Other Comments Or Suggestions:**

None.

**Other Strengths And Weaknesses:**

Strengths

- This paper is well presented and easy to read. Idea is well motivated by prior studies and experiemnts.
- Engineering work will have impacts for future studies.

**Questions For Authors:**

None.

**Relation To Broader Scientific Literature:**

It has a contribution in efficiency in inference, in particular, by employing activation sparsity.

**Theoretical Claims:**

Theoretical analysis is provided in appendix A by contrasting their approach with TEAL (Liu et al., 2025).

---

> ### Author Rebuttal · Authors · 2025-03-29
>
> We sincerely thank the reviewer nBNG for his/her thoughtful feedback and recognition of our work, which has greatly encouraged our research team. Our proposed LaRoSA framework utilizes Computational Invariance Transformation, a well-established technique in model quantization and pruning, to systematically enhance activation sparsity during language model inference. When combined with our carefully designed top-k activation selection mechanism and hardware-optimized custom kernels, this approach consistently accelerates models ranging from 7B to 70B parameters during decoding.
>
> The recent open-sourcing of DeepSeek-R1[1] and subsequent advancements in related research [2-6] have shown that reasoning models using long-chain-of-thought with RL-based test-time scaling make significant progress on complex reasoning tasks. These studies reveal an interesting emergent property: **having more thinking tokens during the decoding phase can significantly extend the capability boundary of the reasoning models**. For instance, DAPO[3] reports that RL training increases the mean response length from 1,000 to 4,000 tokens. The notable increase in output length during the decoding phase offers significant opportunities for optimization, and our LaRoSA method is valuable because it speeds up these lengthy decoding processes without sacrificing model performance.
>
> To further validate LaRoSA's effectiveness, we conducted rapid empirical evaluations on distilled reasoning models, including DeepSeek-R1-Distill-Llama3-8B and DeepSeek-R1-Distill-Qwen2.5-7B[1]. As discussed in our response to Reviewer V3DZ, preliminary results show that **LaRoSA achieves consistent inference speed up while preserving the ability to solve complex reasoning tasks**. Additional ablation studies with Reviewer jWLg **demonstrate compatibility and effectiveness when integrating LaRoSA with semi-structured pruning techniques such as Wanda[7] and RIA[8], further confirming our method's robustness.**
>
> Once again, we deeply appreciate reviewer nBNG's constructive comment on our work. Our team remains committed to refining this research area and will open-source both the implementation code and optimized kernels to encourage community adoption and further development.
>
> [1] Deepseek-r1: Incentivizing reasoning capability in llms via reinforcement learning. arXiv preprint arXiv:2501.12948, 2025.
> [2] Demystifying Long Chain-of-Thought Reasoning in LLMs. arXiv preprint arXiv:2502.03373, 2025.
> [3] DAPO: An Open-Source LLM Reinforcement Learning System at Scale. arXiv preprint arXiv:2503.14476, 2025.
> [4] Open-reasoner-zero: An open source approach to scaling reinforcement learning on the base model, 2025.
> [5] Understanding R1-Zero-Like Training: A Critical Perspective. arXiv preprint arXiv:2503.20783, 2025.
> [6] SimpleRL-Zoo: Investigating and Taming Zero Reinforcement Learning for Open Base Models in the Wild. arXiv preprint arXiv:2503.18892, 2025.
> [7] A Simple and Effective Pruning Approach for Large Language Models, ICLR 2024.
> [8] Plug-and-Play: An Efficient Post-training Pruning Method for Large Language Models, ICLR 2024.

---

### Decision · Program_Chairs · 2025-05-01

**Decision:**

Accept (poster)

**Comment:**

All reviewers recommended accepting the paper.

I myself enjoyed the crisp presentation and clarity of the ideas. The approach is sound and the experimental results are comprehensive. A comprehensive rebuttal of concerns raised by Reviewer V3DZ was given; I encourage the author to update the paper with the results of this discussion, because I believe it will strengthen the paper. The same goes for the new experiments included in the response to Reviewer jWLg.